# PRD: Peer Rank and Discussion Improve Large Language Model based Evaluations

**Ruosen Li**   **Teerth Patel**   **Xinya Du**

*Department of Computer Science, The University of Texas at Dallas*
*{ruosen.li, teerth.patel, xinya.du}@utdallas.edu*

**Reviewed on OpenReview:** `https://openreview.net/forum?id=YVD1QqWRaj`

## Abstract

Nowadays, the quality of responses generated by different modern large language models (LLMs) is hard to evaluate and compare automatically. Recent studies suggest and predominantly use LLMs for reference-free evaluation of open-ended question answering. More specifically, they use the recognized "strongest" LLM as the evaluator, which conducts pairwise comparisons of candidate models' answers and provides a ranking score. However, this intuitive method has multiple problems, such as bringing in self-enhancement (favoring its own answers) and positional bias. We draw insights and lessons from the educational domain (Cho & MacArthur, 2011; Walsh, 2014) to improve LLM-based evaluations. Specifically, we propose (1) the peer rank (PR) algorithm that takes into account each peer LLM's pairwise preferences of all answer pairs, and outputs a final ranking of models; and (2) peer discussion (PD), where we prompt two LLMs to discuss and try to reach a mutual agreement on the preferences of two answers. We conduct experiments on two benchmark datasets. We find that our approaches achieve higher accuracy and align better with human judgments. Interestingly, PR can induce a relatively accurate self-ranking of models under the anonymous setting, where each model's name is unrevealed. Our work provides space to explore evaluating models that are hard to compare for humans.

## 1 Introduction

With a rising number of large language models (LLMs) being developed ever more quickly recently, evaluations become increasingly important as they encode values and priorities that the LLM community should improve upon (Jones & Galliers, 1995; Liang et al., 2022). At the same time, the evaluation becomes harder as well. For example, recent models finetuned with reinforcement learning from human feedback (RLHF) demonstrate greater alignment with human preferences, but this capability usually cannot be reflected by decent performance on standard NLP benchmarks (e.g., MMLU (Hendrycks et al., 2020) and ARC (Clark et al., 2018)). Furthermore, human queries span a diverse range of settings and scenarios, making it nearly impossible to list them all Fan et al. (2019); Ouyang et al. (2022).

To tackle this discrepancy, open-ended questions are being used more often to test LLMs' performance (Chiang et al., 2023). Then, by default, evaluation is done by collecting human preferences of pairwise comparisons and then calculating scores for each LLM to induce a general ranking. Yet the collection process is costly and time-consuming, to automate and scale up the evaluation, most recent works utilize the state-of-the-art LLM as the judge (Dubois et al., 2023; Lin & Chen, 2023). However, various studies show that this method is problematic, as the pairwise comparison judgment provided usually contains various biases, such as favoring LLMs' own answers Liu et al. (2023); Zheng et al. (2023).

Motivated by these limitations, we propose the idea of *peer evaluation*. The goal is to mitigate the biases in automated evaluations while still benefiting from LLM's strong capability in reading and writing reviews. We

propose **P**eer **R**ank and **D**iscussion-based evaluation framework (PRD). The suit consists of two alternatives that share the same format and goal – involving peer LLMs' participation as reviewers to reach a more fair evaluation result where all peers mutually agree. We draw insights and lessons from educational psychology research field on methodologies of student peer reviewing (Walsh, 2014), as well as their impact and benefits (Cho & MacArthur, 2011; Yalch et al., 2019). More specifically, *Peer Rank (PR)* is utilized for global rankings and induces reviewer weights. It works for the tournament-style benchmarking setting where each LLM in pairwise matches produces an answer for an open-ended question. Instead of using the average vote to decide the final preference scoring, we propose weighted votes based on LLMs reviewers' capabilities. *Peer Discussion (PD)* facilitates fine-grained pairwise comparison/ranking. It works for the general pairwise comparison setting. Given two candidate answers, we prompt two other reviewer LLMs to have multi-turn discussions to reach a mutual agreement on the pairwise scoring or preference. The process shares a similar format of LLM interacting with each other through conversations like two communicative agents (Li et al., 2023; Park et al., 2023; Fu et al., 2023b). *PR* and *PD* are closely interrelated and fall under the same theme of providing a more fair (de-biased) ranking of long- and free-form answers.

We conduct extensive experiments and analysis for measuring PR and PD's capabilities of providing fair pairwise comparisons. PR is tested on Vicuna80 Chiang et al. (2023), which contains pairwise judgments from human annotators. Our method improves correlations with human rankings substantially. This paradigm also enables a group of LLMs to induce a self-ranking. PD is tested on both Vicuna80 and LFQA (Xu et al., 2023), which includes annotated pairwise comparisons of Human-Machine and Machine-Machine answers. PD enables LLMs to achieve better pairwise comparisons that are more accurate than single model-based reviews. Both PR and PD mitigate the above mention biases especially self-enhancement bias significantly. Further, we provide more analysis for PD, showing: (1) the LLM leading discussions is less likely to alter its opinion; (2) stronger LLMs are more likely to hold their opinions.

## 2 Related Work

**Automatic Evaluations** Natural Language Generation (NLG) evaluation methods are mainly of a similarity-based or reference-free type. For similarity-based metrics, the generated texts are compared to reference texts Papineni et al. (2002); Zhang et al. (2019). In parallel, people have also developed task-specific metrics such as consistency (Kryściński et al., 2020; Wang et al., 2020), faithfulness (Fabbri et al., 2022; Gao et al., 2023) and coherence (Dziri et al., 2019). This is similar to our peer discussion idea on designing more specific prompts for large language model-based evaluations. Our prompting-based method is more flexible and can act as a unified evaluator (Zhong et al., 2022).

Specifically, for **long-form or open-ended question answering**, early work uses ROUGE (Lin (2004)) to measure the similarity between human and machine-generated answers. However, researchers find that ROUGE is not a fair metric for quality measurement due to the open-ended nature of long-form answers (Krishna et al., 2021; Xu et al., 2023). Fu et al. (2023a) propose GPTScore, which evaluates texts with generative pre-training models like GPT-3. Xu et al. (2023) implements a similar idea for evaluating long-form answers. Given a prompt consisting of a question with two answer candidates, GPT-3 is fine-tuned to output the label answer 1 or answer 2 (**pairwise comparisons**).

**LLMs as evaluators: problems and challenges** Most recently, with the trend of developing more LLMs, evaluations for benchmarking the progress have become even more important but also more difficult. They are tested on both standard datasets such as MMLU, and more importantly, on open-ended questions which are much more prevalent in real life (Nakano et al., 2021; Chiang et al., 2023). People mostly use GPT-4 (Liu et al., 2023; OpenAI, 2023) as an evaluator for either generating scores or pairwise comparisons (Wang et al., 2023b; Zhou et al., 2023). However, such a strategy has fundamental problems because of various biases, such as (1) positional bias (Dettmers et al., 2023; Wang et al., 2023a), where a model favors the first answer in pairwise comparisons; (2) verbosity and length bias (Wang et al., 2023b); (3) and most importantly, self-enhancement bias, where an LLM favors its own answers (Liu et al., 2023; Zheng et al., 2023).

Efforts have been proposed to tackle them: (1) Using position switching (Wang et al., 2023a) for mitigating positional bias; (2) Zheng et al. (2023) proposes Chatbot Arena, where real users ask questions and provide

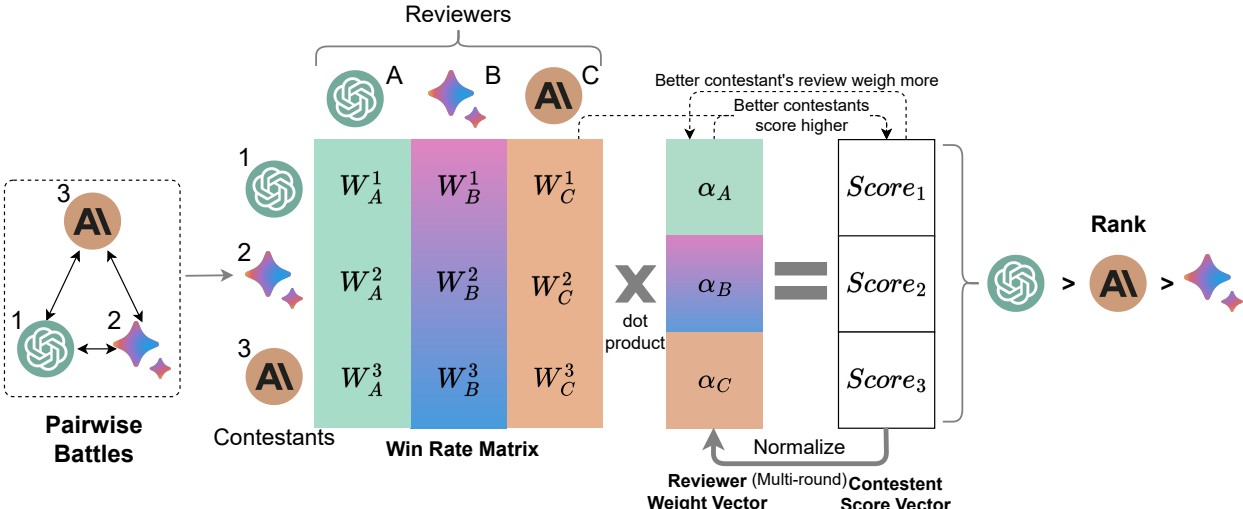

Figure 1: The peer rank process (PR): each LLM model acts both as reviewers (A, B, C) and contestants (1, 2, 3). From the battles between contestants (pairwise comparisons), it induces a self-ranking. In this example, models A, B, and C represent GPT-4, Bard, and Claude, respectively.

pairwise judgments of answers generated by two LLMs. But this is time-consuming and requires expert annotation to ensure fairness; (3) Bai et al. (2023) propose using each LLM as an examiner, where each generates questions to test other models. Different from PD, their "exams" are biased with randomly generated questions. Moreover, none of the above works support inducing self-rankings through peer ranking.

Overall, our work, peer evaluation-oriented methods, undergo tests on two benchmarks, each covering a variety of tasks, and focuses more on the alignment between LLMs' evaluations and human judgments.

**LLM Interactions and Discussion**  There have been works that explore Multi-agent discussion with LLMs, including Liang et al. (2023); Du et al. (2023); Chan et al. (2023); Chen et al. (2023). All of them are task-oriented frameworks aiming at improving LLMs' performance on general tasks. Du et al. (2023) focused on math tasks and question answering tasks. Liang et al. (2023) tested two tasks, including machine translation and question answering. Chen et al. (2023) covered math and reasoning. Chan et al. (2023) implemented a multi-agent debate framework with multiple prompts for evaluation. Prior works utilize LLM interactions to accomplish tasks and improve models' accuracy. For our case, based on one response from humans and another from LLM, our approach utilizes LLM interactions to discuss which one is better and in order to achieve better evaluation that align with human preferences.

**Peer Evaluation in the Educational Domain**  Prior works on educational research mainly focus on human-in-the-loop studies, such as in the classroom (Cho & MacArthur, 2011; Nicol et al., 2014). They conduct human-oriented data collection and experiments to verify the benefits of peer evaluations (Walsh, 2014). In contrast, we focus on automatic evaluation, employing peer LLM reviewers to conduct pairwise comparisons of LLMs' answers. Moreover, our peer rank process focuses on pairwise comparisons, instead of absolute grades.

## 3  Methodologies

In general, Peer Rank (PR) can be applied to induce self-ranking – a ranking of a group of LLMs' own capabilities. Peer Discussion (PD) provides a more fine-grained and interactive comparison of two models' answers. Both of them aim at reducing the bias in automatic evaluations. We elaborate on the technical details in this section.

### 3.1 Peer Rank and Scoring (PR)

Figure 1 illustrates the peer rank algorithm. The general idea is to obtain weighted scores of each battle from the peer reviewer's judgment, and then induce self-rankings from the scores. This process is iterated multiple times until the scores converge.

Given a set of questions $Q$, we generate an answer to each question from each LLM. Let $A_m(q)$ be the answer to question $q \in Q$ by the model $m$. Each *battle* represents two models (the contestants) answering the same question $q$. The comparison of the answers in a battle by the LLM reviewer model $r$ forms a *review*. Let $K_r(x, y)$ be the score given by the reviewer $r$ to the pair of answers $(x, y)$. We use a score of $-1$ to indicate the first answer is better, 0 to indicate a tie, and 1 to indicate the second answer is better. Suppose we have a set of reviewer models $R$ and a set of contestant models $C$. We form a set of battle reviews, $B = \{(q, i, j, r, s) \mid q \in Q, (i, j) \in C^2, r \in R\}$, where $s = K_r(A_i(q), A_j(q))$ is the score given by reviewer $r$ to the answers/responses generated by $i$ and $j$ for question $q$. We create a shorthand $K_r^{ij}(q)$ for this review.

Based on these peer reviews, we can evaluate models based on their performance by calculating metrics such as the win rate of each contestant and the Elo ratings (Section 3.1.2) of each contestant. Since each model is ranked by its peers, we call it Peer Rank.

Specifically, the set of questions should be diverse and cover various tasks, such as question answering (12.5%), email writing (12.5%), coding (6%), math solving (4%), etc. Answers/responses should also vary in format, including concise answers, step-by-step reasonings, detailed explanations, code snippets, long-form answers, etc. Reviewers assess response pairs and indicate preferences in the process ("battle review"). Then, both winrate and Elo metrics can be calculated.

#### 3.1.1 Win rate Calculation

The win rate for a contestant is the number of wins for that contestant divided by the number of battles it participates in. Ties are counted as 0.5 wins for both contestants.

Our win rate calculation assigns differing weight to the scores provided by different reviewers (A, B, C) based on the performance of the corresponding reviewers as a contestant (1, 2, 3). This operates on the assumption that models which are better contestants are also more fit to evaluate and compare answers, so they should be given more weight in evaluation (Equation 2). In another way, *since the score is a measure of their ability to review/grade correctly, we weigh the win rate an LLM gives another LLM by their own score* Walsh (2014). Moreover, the self-rewarding paper by Yuan et al. (2024) also made this assumption. They use a model itself for evaluations during the iterations and prove that it makes sense. In their results, the better-performing models also perform well in providing high-quality evaluations/rewards to themselves.

Initially, all reviewers are given the same weight. On each iteration of the calculation, the win rate for each contestant is calculated using the current weights. The win rates are scaled to the range of $[0, 1]$ using a linear scaling. Then, they are scaled again so that their sum is 1. Next, these results are used as the weights for the next round.

Formally, let $W_r^c$ be the raw win rate of contestant $c \in C$ from the reviews of reviewer $r \in R$. This is equal to the number of times $c$ wins a battle plus half of the number of times $c$ ties, divided by the number of battles $c$ participates in.

$$W_r^c = \frac{\sum_q \sum_{d \in C, d \neq c} \left[ f(K_r^{dc}(q)) + f(-K_r^{cd}(q)) \right]}{2|Q|(|C| - 1)} \tag{1}$$

where $f(score) = \frac{score + 1}{2}$ maps a score of (loss $= -1$, tie $= 0$, win $= 1$) for the second contestant to a win count of (0, 0.5, 1), so that ties count as half of a win.

Note that we negate $K_r^{cd}(q)$ when inputting it into $f$ so that the win value of $c$ is computed instead of $d$. Also, since there are $|Q|$ questions, $|C - 1|$ contestants to battle, and 2 orders for two contestants to battle, there are $2|Q||C - 1|$ battles involving a fixed contestant $c$.

Let $\alpha_r^k$ be the weight assigned to reviewer $r$ after iteration $k$. Initially, $\alpha_r^0 = 1/|R|$, so that all reviewers have the same weight, and the weights add to 1. Namely, we assume each reviewer LLM has the same capabilities to start. The score of contestant $c \in C$ for iteration $k$ is the weighted average of the raw win rates for contestant $c$. We set the weights for the next iteration to $\alpha^k$:

$$\text{score}_c^k = \sum_{r \in R} \alpha_r^{k-1} \cdot W_r^c, \alpha^k = \text{Norm}(\text{MinMax}(\text{score}^k)) \tag{2}$$

where the weights are scaled to a range of $[0,1]$ and finally normalized to have sum equal to 1:

$$\text{MinMax}(S) = \frac{S - \min_{r \in R}(S_r)}{\max_{r \in R}(S_r) - \min_{r \in R}(S_r)}$$

Given this set of equations, we look for the fixed/converging point of the framework. This process is reminiscent of the problem faced by the PageRank algorithm (Page et al., 1999). The detailed equivalent implementation of PR is shown in the Algorithm 2 in Appendix E.

The whole process is simple but effective. It automatically adjusts weights for all models and mitigates the self-enhancement bias. During the process, every reviewer model's score is considered instead of only one model itself (Equation 2). While a reviewer may favor its outputs, other reviewers' scores provide a fair balance. Additionally, weaker models' reviewing weights decrease automatically (to near zero) because of the normalization operation. Empirical tests showed that fixing the self-weight at zero resulted in poorer performance.

### 3.1.2 Elo Calculation

Another method for calculating the performance of a contestant relative to other contestants is the Elo rating (Elo, 1967; Askell et al., 2021), which is utilized for ranking players and widely used in games (battles) (Dettmers et al., 2023). It measures the relative skill levels of players by predicting the expected win rate against opponents. It takes a sequence of pairwise reviews and generates ratings for each contestant, with a greater rating indicating better performance. This is a more fine-grained measurement compared to win rate. Based on a similar idea, we assign different weights to reviewers based on their previous performance such that a review from a higher-weight reviewer has a greater influence upon Elo ratings.

Similarly to the win rates calculation, we start with equal weights on all reviewers and then normalize the resulting Elo ratings to give weights for the next iteration. We repeat the Elo calculation with the new weights, update the weights based on the new ratings, and continue repeating until it converges. Walsh (2014) has proved the guarantee of convergence. Compared to Walsh (2014), which involves student grading in the educational domain, our work focuses on peer LLM reviewers conducting pairwise comparisons of LLMs' answers, utilizing winrate and Elo scores. We introduce an automatic LLM peer evaluation metric for the machine learning field and extend Walsh's convergence proof, showing our method's reliability through experimental results. [1] More difference details are in appendix C.

A brief overview of the actual Elo ratings calculation follows. All contestants start out with an initial rating of 1000. On each battle, the expected likelihood of each contestant winning is calculated based on the difference between their Elo ratings. The Elo rating of the winner is increased, and the rating of the loser is decreased. The magnitude of the Elo ratings change is inversely related to the outcome's likelihood. In our calculations, we weigh reviewers so that reviews by a high-weight reviewer cause larger changes in Elo. For more details, please refer to Algorithm 1 in Appendix E.

### 3.2 Peer Discussions (PD)

In peer discussion, we prompt two LLMs to discuss how to judge two candidate answers, trying to reach a final agreed review. All prompts include detailed instructions and specify the output format for the LLMs. In Figure 2, we demonstrate the peer discussion process between two LLMs reviewers (A and B). The input is a given question and two answers, which may be both generated by machines or one by humans and another by

---

[1]We provide the proof of convergence of our peer rank algorithm in Appendix A.

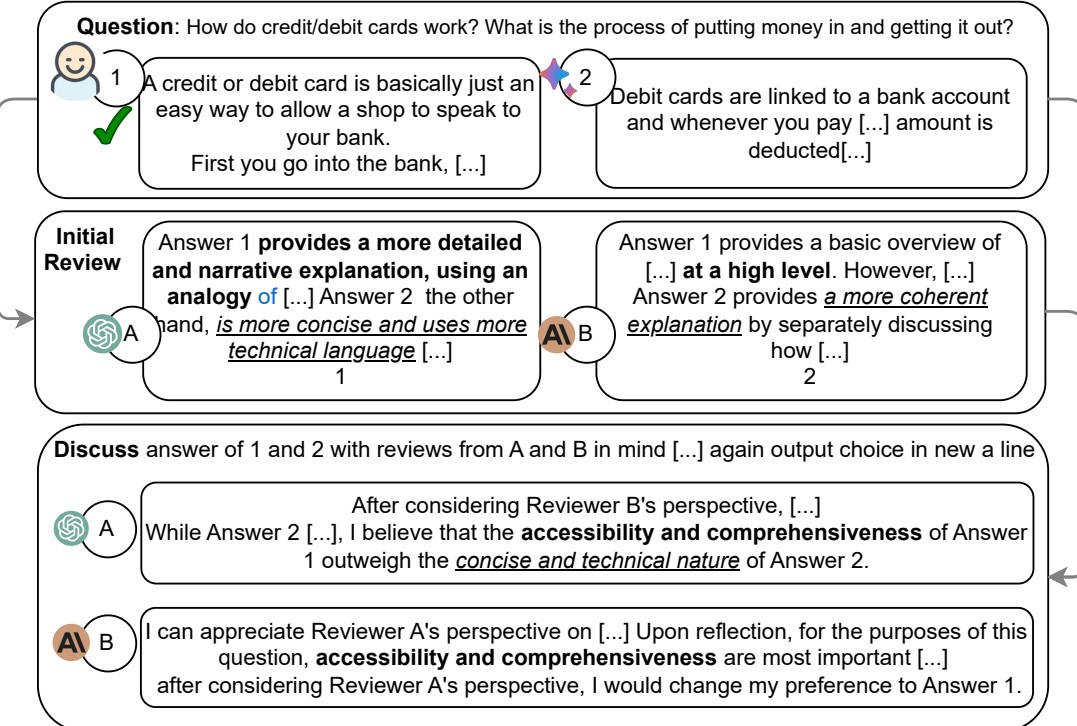

Figure 2: The peer discussion process (PD). **Bold** and _italic_ texts describe the advantages of answer 1 and answer 2. In this example, finally, the two LLM reviewers reach the mutual agreement of selecting answer 1 (human-written answer), which correlates with human annotator preference.

```
[System] You are a helpful and precise assistant for checking the quality of the answer.
[Question] {Q}
[Answer1] {A1}
[Answer2] {A2}
[System] We would like to request your feedback on the performance of two answers in
response to the user question displayed above.
Firstly, please compare the two answers based on if they contain unsupported    informa-
tion, core information, and coherence.  Please provide a comprehensive explanation of your
evaluation, avoiding any potential bias and ensuring that the order in which the responses
were presented does not affect your judgment.
Once you have carefully reviewed both submissions, in a new line, choose between the two
answers by outputting the number 1 or 2 respectively.  Do not output anything else other
than the number in this last line.
```

Table 1: It shows the review template for reviewers with three slots ({Q}, {A1}, and {A2}). We instruct reviewer models to focus on **core aspects** , whose definitions are in appendix F. As mentioned in Wang et al. (2023a), position bias still exists after _emphasizing it_ in the prompt.

machines (e.g. GPT-3 v.s. human answers). They first conduct pairwise comparisons on answers separately, providing explanations and indicating their preferred answer by outputting the number 1 or 2 by the end (the prompt for getting initial reviews is listed in Table 1). Then, the two models discuss multiple turns until they reach a fixed number of turns.

The specific prompt for discussion is listed in Table 2. At the very beginning, a system prompt (role prompt) tells models their role – whether it is reviewer A or reviewer B (e.g., Claude or GPT–4). Then, all information, including the question, two comparison answers, and the initial reviews, are listed line by line. The order of initial reviews is the same as that of reviewers in discussions. In other words, if reviewer A leads the

```
[System] You are reviewer A, discussing with reviewer B about your reviews of the following
answers.
[Question] {Q}
[Answer1] {A1} [Answer2] {A2}
[Init Review A] {Review of reviewer A} [Init Review B] {Review of reviewer B}
[System] "Read the reviews and discussions above, and make a decision if to change your
preference, and explain.  Remember we focus on unsupported information, core information,
and coherence.  In a new line, choose between answer 1 and answer 2 by outputting the number
1 or 2 respectively.  Do not output anything else other than the number in this last line."
[Reviewer A] {First-turn output}
[Reviewer B] {Second-turn output}
[Reviewer A]:
```

Table 2: The discussion template for reviewer A at the third turn. Similar to the review template, we explicitly indicate **aspects** that reviewers need to pay attention to. All texts above are chat history which are input to reviewer A LLM model.

discussion, reviewer A's initial review is listed first. Right before the start of the discussion, the system prompt specifies the detailed requirements, which provide explicit aspects to focus on.

Specifically, we draw insights from WebGPT (Nakano et al. (2021))'s annotation guideline OpenAI (2022). For long-form question answering, we focus on (1) *Unsupported information*: detecting information with no support, assume the worst case: that all of it is false. This aspect is most important and often determines the overall rating; (2) *Core information*: about whether the question has actually been answered; (3) *Coherence*: generally, it is less important than the two above. Then, the overall preference is finally determined. Moreover, we do not adjust the above prompts based on any datasets we test in experiments.

## 4  Experiments and Analysis

### 4.1  Datasets, Metrics, and Setup

#### 4.1.1  Datasets

We select two "meta-evaluation" datasets, LFQA (Xu et al., 2023) and Vicuna80, with human annotations for pairwise comparisons, to measure the correlation between our evaluation methods and human judgments.

**LFQA** (Xu et al., 2023) contains 140 long-form questions across seven domains (e.g., economics, history, and biology) and two candidate answers (from either GPT3 or Human) for each. Similar to ELI5 (Fan et al., 2019), it contains more recent (i.e., after July 2021) questions from Reddit forums "r/explainlikeimfive" and "r/AskHistorians". The authors collected expert-level annotations of which answer is better (overall preference).

**Vicuna80** (Chiang et al., 2023) is a set of 80 open-ended questions, which spans 9 categories and covers a wide range of tasks, including question answering, email writing, math problems, etc. In the QLoRA work (Dettmers et al., 2023), authors annotated pairwise comparison scores (overall preference) across 7 models for each question. The scores include 0, 1, 2, which correspond to tie, model 1 wins, and model 2 wins respectively. We select pairwise comparison annotations of 4 models' answers (i.e., GPT4, ChatGPT-3.5., PaLM-2, Vicuna-13b). To make our study more comprehensive, we add recent proprietary language models such as Claude. Specifically, we also annotate pairwise comparisons between Claude's answers and the other 4 models'. We term this a more complete version of the dataset Vicuna80. More details about the annotation process are provided in Appendix I. Since answers to open-ended questions are even harder to compare, the annotators achieve a fair agreement.

**SummEval** (Fabbri et al., 2020) is a benchmark evaluating summarizations. It contains 1600 summaries over 100 news articles from CNN/Daily Mail dataset (Hermann et al., 2015). Each summary is evaluated in four aspects: `coherence`, `consistency`, `fluency`, and `relevance`. In our experiments, we take the average of the four metrics as the evaluation results.

| models | GPT-4 | | All | | All (Weighted) | | Human Raters | |
|---|---|---|---|---|---|---|---|---|
| | Elo | Rank | Elo | Rank | Elo | Rank | Elo | Rank |
| GPT-4 | 1282 | 1 | 1165 | 1 | **1213** **(-23)** | 1 | 1236 | 1 |
| Claude | 1150 | 2 | 1104 | 2 | **1125** **(-2)** | 2 | 1127 | 2 |
| Vicuna | 883 | 4 | 930 | 3 | **912** **(-8)** | 3 | 920 | 3 |
| GPT-3.5 | **890** **(+22)** | 3 | 919 | 4 | 894 | 4 | 868 | 4 |
| PaLM-2 | 804 | 5 | 881 | 5 | **856** **(+8)** | 5 | 847 | 5 |

| models | GPT-4 | | All | | All (Weighted) | | Human Raters | |
|---|---|---|---|---|---|---|---|---|
| | Win Rate | Rank | Win Rate | Rank | Win Rate | Rank | Win Rate | Rank |
| GPT-4 | 0.856 | 1 | 0.749 | 1 | **0.802** **(-0.020)** | 1 | 0.822 | 1 |
| Claude | 0.709 | 2 | 0.662 | 2 | **0.685** **(-0.004)** | 2 | 0.689 | 2 |
| Vicuna | 0.340 | 4 | **0.393** **(+0.004)** | 3 | 0.376 | 3 | 0.389 | 3 |
| GPT-3.5 | **0.342** **(+0.028)** | 3 | 0.375 | 4 | 0.346 | 4 | 0.314 | 4 |
| PaLM-2 | 0.245 | 5 | 0.320 | 5 | **0.290** **(+0.004)** | 5 | 0.286 | 5 |

Table 3: The above tables show results performed on the Vicuna80 dataset. The rows represent the contestants in battles, and the columns represent evaluation methods. The upper table shows the correlation of Elo scores between LLM reviewers and human rater. Bottom table shows the correlation between global win rates. Additional results on emerging (more) LLMs in Table 4 further verify the consistent robustness of PR. **Boldfaced numbers** are the closest to scores from human raters. **Blue numbers** show the difference between the scores from LLM reviewers and Human raters. For more detailed pairwise win rates, please refer to the heat maps in Section 4.2.

| models | All | | All (Weighted) | | Human Raters | |
|---|---|---|---|---|---|---|
| | Elo | Rank | Elo | Rank | Elo | Rank |
| Vicuna | 999 | 2 | 1011 (-47) | 1 | 1058 | 1 |
| Zephyr | 1010 | 1 | 1007 (+7) | 2 | 1000 | 2 |
| GPT-3.5 | 991 | 3 | 993 (+52) | 3 | 941 | 3 |

Table 4: Additional Elo Scores for Vicuna, Zephyr and GPT-3.5

In LFQA, questions receive 1-3 expert-level annotations per category, with human agreement ranging from 0.4 to 0.65. Each Vicuna80 question receives 3 human annotations, with human agreement between 0.5 and 0.62. Each SummEval summary is annotated by 8 humans, with an agreement number of 0.7. We use the human majority vote as the human preference during battles.

### 4.1.2 Metrics

For experiments on PR, we follow metrics in Wang et al. (2023a). We first conduct *example-level* pairwise comparisons. Specifically, each evaluation example (pairwise comparison) consists of a question and a pair of answers. We compare the model predicted preference score against gold human preference and report Accuracy and Fleiss' $\kappa$, a statistic that measures the reliability of agreement among multiple raters. Specifically, Fleiss' $\kappa$ is employed to gauge alignment between the model's predictions and human preferences, where a higher score signifies a stronger alignment. Following Dettmers et al. (2023), we also compare model-predicted global ranking scores against human-judged ranking scores. Specifically, we report Elo scores (Elo, 1967) and win rate (WR) based rankings (Table 3). We use **All** to denote our method where each reviewer has equal weights, and use **All (Weighted)** to denote the setting where the final round weights are applied to each reviewer. Besides experiments on PR and PD respectively, we also compare PR and PD in an experiment of judging answer qualities of GPT-3.5 v.s. Vicuna-13b (Section 4.2 and 4.3).

For experiments on PD, we use peer discussion accuracy (PDA) to describe the correlation of model discussion results compared to human annotators. PDA is calculated by the number of correct answers from peer discussion results over the number of all answers. A high PDA result indicates a better correlation with human judgments.

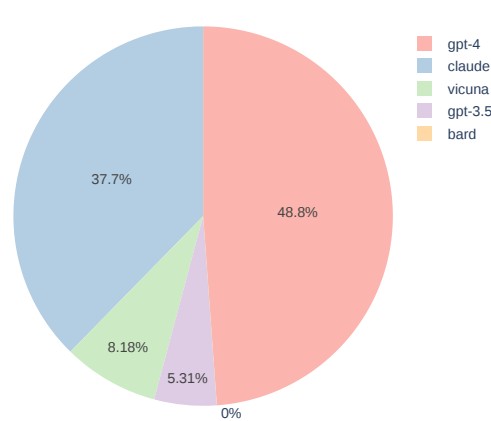

Figure 3: Peer rank final round weights of each LLM reviewer. GPT-4 and Claude take 48.8% and 37.7% weights. Bard got close to zero weights.

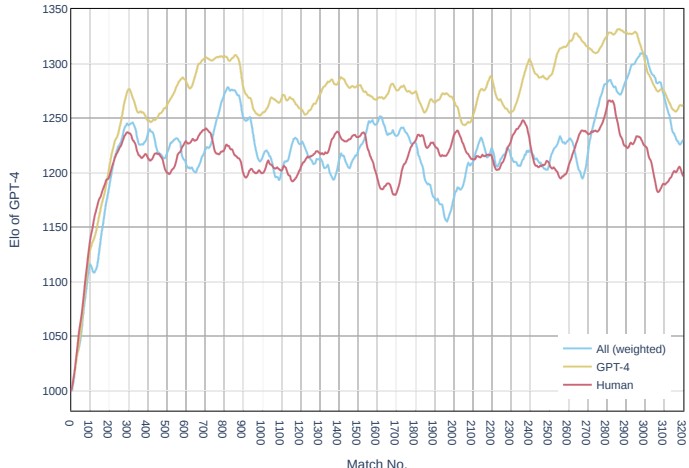

Figure 4: GPT-4 Elo scores every 100 battles on the Vicuna80. Elo scores provided by the GPT-4 reviewer are consistently higher than human ratings, while our All (weighted) ratings correlate with humans well.

### 4.1.3 Setup

For Vicuna-13b, we use the default version from Chiang et al. (2023). For all other API-based LLM models, we use specific versions of each, i.e., `GPT-4-0613`, `GPT-3.5-turbo-0613`, `Claude-1`, and `Text-Bison@001` for GPT-4, GPT-3.5, Claude, and PaLM-2 respectively. For more details, please refer to appendix B. For discussions in the PD method, we set the maximum number of turns as 4. Based on our experiments, most discussions reach mutual agreements at turn 4. Moreover, the default temperature for all models is 0.2.

## 4.2 Results for Peer Rank (PR)

On the Vicuna80 dataset, we compare our PR method and representative LLM-based evaluation methods, such as GPT-4 and Claude.

In Table 5, all reviewer combinations listed except Claude, when compared to human reviews at an example level, display a Fleiss' $\kappa$ of around 0.40, indicating fair to moderate agreement. There is a significant difference in accuracy between LLM reviewers. The worst reviewer is Claude, with an accuracy of only 60.7%. The best individual reviewer is GPT-4, with an accuracy of 64.3%. The combination of reviewers (PR) increases this accuracy by a few percentage points, with our PR approach being highest at 67.3%.

| Reviewer | Fleiss' $\kappa$ | Accuracy |
|---|---|---|
| GPT-3.5 | 0.387 | 0.621 |
| Claude | 0.319 | 0.607 |
| GPT-4 | 0.406 | 0.643 |
| GPT-4 & Claude & GPT-3.5 | 0.403 | 0.666* |
| All Reviewers (Weighted) | 0.410 | **0.673**** |

Table 5: Example-level correlation results, for the fourth and fifth rows, we take the peer reviewers' majority vote weighted by winrate. Two-tailed t-test results statistical significance is indicated with *($p < 0.01$), **($p < 0.002$).

Inspecting Table 3, GPT-4 reviewer ranks GPT-3.5 higher, while our All (Weighted) achieves the same ranking as humans: i.e. `GPT-4 > Claude > Vicuna > GPT-3.5 > PaLM-2`. This shows that a weighted peer ranking provides a more accurate evaluation of the global performance of models. Moreover, the ranking corresponds to that in the Chatbot Arena Leaderboard[2]. Thus, the assumption in section 3.1.1 can be verified. In terms of the Elo ratings provided by the human reviews, we clearly observe that GPT-4 clearly favors its own answers and is prone to *self-enhancement* bias. Our approach (All (Weighted)) produces the closest Elo ratings. Furthermore, it also produces the closest win rates (less than a 1% difference for many contestants). In the beginning, when the weight is the same for every reviewer (weights equal to one), the win rate given by "All reviewers" is low at about 0.749 partially because each reviewer is treated equally so that each reviewer might have a preference for its own answer. After several rounds/iterations, the final win rate becomes more fair. We display the final round weights in Figure 3.

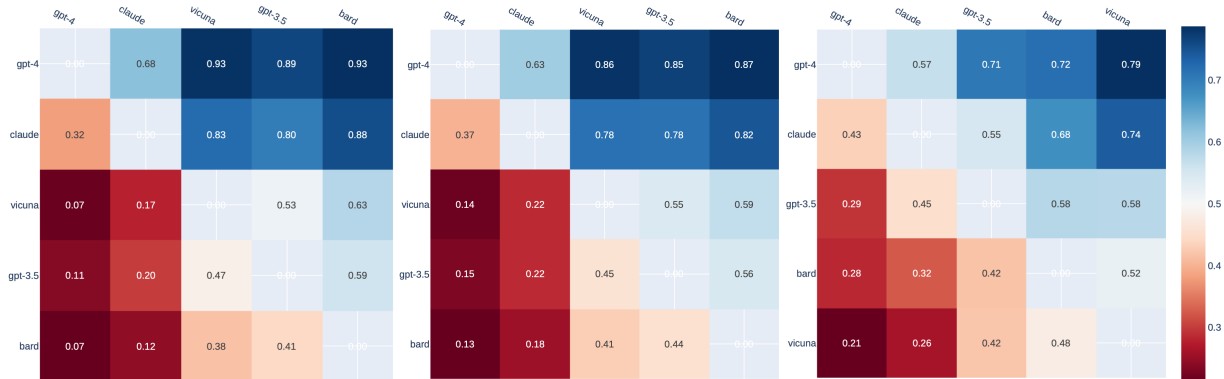

Figure 5: Pairwise win rate heatmaps: Fraction of Model A Wins for all A vs. B Battles (A: rows, B: columns). Left: GPT-4 evaluator; Middle: our method All (weighted); Right: Chatbot Arena pairwise win rate. All results in the three sub-figures are generated separately using the same data.

In Figure 4, we draw the line chart of how the GPT-4 Elo score changes as more battles are fed to the Elo algorithm. GPT-4's score takes off as battle number increases. We can observe that GPT-4 displays self-enhancement across the entire process, while our PR-based evaluation correlates with human pairwise comparisons well.

In Figure 5, we present the detailed pairwise win rates between every two contestants (LLMs). We compare our evaluation with GPT-4 based evaluation, as well as the Chatbot Arena leaderboard. The Arena ranking[2] is based on user queries and their corresponding preferences for two responses. The figure demonstrates that although both approaches favor GPT-4 and Claude answers, the win rates calculated by our approach All (weighted) correlate better with Arena win rate, especially on weaker models. More pairwise win rate heatmaps are in Appendix H.

## 4.3 Results for Peer Discussions (PD)

**Prompt for Discussion** By preliminary study, we find that the template asking for explicit aspects, such as core information, unsupported information, and coherence, can substantially help LLM reviewers generate valuable and informative reviews which correlate better with human annotators.

We first conduct preliminary experiments to find a relatively

|  | GPT-4 lead | Claude lead | Random |
|---|---|---|---|
| GPT-4 init score | - | - | 0.729 (±0.014) |
| Claude init score | - | - | 0.671 (±0.025) |
| Generic prompt | 0.714 (±0.018) | 0.671 (±0.022) | 0.686 (±0.022) |
| w/ explicit criteria | 0.729 (±0.014) | 0.721 (±0.018) | 0.720 (±0.014) |
| w/ role | 0.743 (±0.011) | 0.729 (±0.018) | 0.729 (±0.011) |
| w/ role & explicit criteria | 0.750 (±0.014) | 0.721 (±0.011) | 0.743 (±0.011) |

Table 6: Different prompting's effect on Peer Discussion Accuracy (on the LFQA dataset). The first two rows are the results before discussions (from GPT-4 and Claude respectively). The last three rows are results after discussions.

good prompt for facilitating LLM peer discussions. The first two rows in Table 6 lists the Peer Discussion Accuracy (PDA) of GPT-4 and Claude's initial pairwise comparison preference before discussions. They have a moderate agreement with human preference, with GPT-4 leading around 5%. For the discussion-based evaluators, we report three types. By "GPT-4 lead", we refer to the discussions where GPT-4 first expresses opinions; by "random", we refer to discussions where the leader is randomly picked.

In discussions (the last three rows), when we use a generic prompt (such as "pick your preferred answer"), the discussion's final preference PDA is around 0.69, higher than Claude's initial judgment's PDA but lower than GPT-4's. When we add more explicit aspects into the prompt [3], the PDA boosts significantly (4% improvement). When we add the role/identity information (Appendix G) to each turn's prompt ("w/ role")

---

[2] https://lmsys.org/blog/2023-05-25-leaderboard/
[3] We select aspects from WebGPT annotation guidelines mentioned in the previous section.

| | R1 | R2 | R1 lead | R2 lead | Random | Random (Best Prompt) |
|---|---|---|---|---|---|---|
| GPT-4 & Claude | 0.729 (±0.014) | 0.671 (±0.025) | **0.743\*** (±0.011) | 0.729 (±0.018) | 0.729 (±0.011) | 0.740 (±0.011) |
| GPT-4 & GPT-35 | 0.729 (±0.015) | 0.579 (±0.023) | 0.714 (±0.011) | **0.750\*** (±0.018) | 0.731 (±0.014) | 0.736 (±0.014) |
| GPT-35 & Claude | 0.579 (±0.026) | 0.671 (±0.023) | **0.700\*** (±0.018) | 0.671 (±0.014) | 0.686 (±0.014) | 0.693 (±0.014) |
| GPT-35 & GPT35-0.8 | 0.579 (±0.026) | 0.650 (±0.040) | 0.664 (±0.018) | **0.686\*** (±0.031) | 0.681 (±0.020) | 0.686 (±0.014) |
| Claude & Claude-0.8 | 0.664 (±0.022) | **0.707** (±0.034) | 0.693 (±0.018) | 0.671 (±0.027) | 0.680 (±0.026) | 0.693 (±0.014) |
| GPT-4 & GPT-4-0.8 | 0.729 (±0.014) | 0.757 (±0.022) | **0.779\*** (±0.014) | 0.757 (±0.018) | 0.779 (±0.018) | 0.786 (±0.014) |

Table 8: Peer discussion accuracies (PDA) on LFQA. "-0.8" indicates the temperature is 0.8. Statistical significance is indicated with $^*(p < 0.05)$. "Best prompt" indicates the discussion uses the best prompt in table 6. (±Numbers) represent standard deviations.

| R1 vs R2 | R1 | | R2 | | R1 lead | | R2 lead | | Random | |
|---|---|---|---|---|---|---|---|---|---|---|
| | $\rho$ | $\tau$ | $\rho$ | $\tau$ | $\rho$ | $\tau$ | $\rho$ | $\tau$ | $\rho$ | $\tau$ |
| GPT-4 & GPT-35 | 0.293 | 0.233 | 0.262 | 0.251 | **0.297** | **0.266** | **0.284** | **0.219** | 0.292 | 0.233 |
| GPT-35 & GPT-35-0.8 | 0.262 | 0.251 | 0.211 | 0.178 | 0.264 | 0.207 | **0.340** | **0.328** | 0.334 | 0.264 |
| GPT-4 & Claude | 0.293 | 0.233 | 0.234 | 0.200 | **0.344** | 0.268 | **0.335** | **0.282** | 0.341 | 0.268 |

Table 9: Summary-level Spearman ($\rho$) and Kendall-Tau ($\tau$) correlations between model evaluation results and the Summeval Benchmark. Columns R1 and R2 contain the results before discussions. The rest of the columns represent discussion results. Variances of all scores are lower than 0.06. P-values for all discussion results (boldfaced) are less than 0.05.

to remind the reviewer, the PDA scores increase for both models, indicating the role information is helpful for LLMs in discussions.

**General Accuracy** In Table 8, we report the peer discussion accuracy (PDA) of multiple combinations of reviewers' discussion results on LFQA based on the best two discussion prompts in Table 6. We observe: (1) when two reviewer LLMs are of similar capabilities (e.g., GPT-4 and Claude differ by less than 100 in Table 3), they reach stable improvements upon their initial reviews. In the above example, GPT-4 gets 3% improvement (from 0.729 (±0.014) to 0.743 (±0.011)), and Claude get 8% improvement (from 0.671 (±0.025) to 0.729 (±0.018)) ;

| | Accuracy |
|---|---|
| GPT-4 | 0.3500 |
| (PD) GPT-4 & Claude | 0.3675\* |
| (PR) All | 0.4375 |
| (PR) All (weighted) | 0.4625 |

Table 7: Comparing peer discussions (PD) and peer ranking (PR) on Vicuna80 (random order is applied to the GPT4 & Claude discussion).

(2) when there is a substantial gap between reviewer capabilities (e.g., GPT-4 and GPT-35 differ larger than 100 in Table 3), the PDA of the weaker model always reaches the most improvement and the highest final result. In the above example, GPT-35 receives a 30% improvement (from 0.579 (±0.026) to 0.700 (±0.018)). ; (3) when models "self-discuss", for example, we create two variants of the same model by setting different temperatures and prompt them to discuss, weaker models (e.g., GPT-3.5) can substantially "self-improve" (from 0.579 (±0.026) to 0.664 (±0.018)). GPT-4's self-discussion brings little improvement (from 0.729 (±0.014) to 0.779 (±0.014)). Future investigations on how to design better self-discussion strategies would be worth working on.

Table 9 shows the same trend as Table 8. A higher correlation score indicates discussion results are more aligned with human annotations. Models with similar capabilities (GPT-4 & Claude) get large improvements after discussion. Models having a substantial gap (GPT-4 & GPT-35) reach results close to the stronger model. When a model self-discuss (GPT-35), it can improve its own performance.

Table 7 reports accuracy on comparisons of GPT-3.5 v.s. Vicuna-13b answers to Vicuna80 questions, we see the GPT-4 & Claude discussion increases the accuracy by over 1.5%. Also, we compare with PR method and find that the review becomes substantially better after weighted scoring.

| Reviewers | Initial Preference | | After Discussion | |
|---|---|---|---|---|
| | GPT-3 Answer First | Human First | GPT-3 First | Human First |
| Human | 57.89% | 59.46% | 57.89% | 59.46% |
| GPT-3.5 | 73.68% | 59.46% | 67.11% | 58.56% |
| Claude | 63.16% | 64.41% | 55.70% | 55.41% |
| GPT-4 | 54.51% | 56.37% | 58.27% | 58.30% |

Table 11: GPT-3 answer win rate (in the GPT-3 vs Human battles). Position bias is mitigated.

**Peer discussions help mitigate self-enhancement bias**  According to what we previously discovered, LLMs endure self-enhancement bias when acting as judges – preferring the answers they generate or that of the models under the same series (e.g., GPT-4 and GPT-3).

We conduct experiments on the subset of LFQA questions where we have human-annotated pairwise comparisons between Human and Machine-generated (GPT-3 `text-davinci-002`) answers. Table 10 shows the win rates of GPT-3 judged by humans and three LLMs. We report the LLMs' initial and after-discussion preferences. Both GPT-3.5 and Claude highly prefer GPT-3's answers in their initial reviews. Specifically, GPT-3.5 significantly favors GPT-3 answers with a 13.79% higher win rate. After discussing with other LLMs, all models align better with humans. Our peer discussion method largely helps GPT-3.5 mitigate self-enhancement bias. Before discussions, GPT-4's

| Reviewers | GPT-3 | |
|---|---|---|
| | Initial Preference | After Discussion |
| Human | 58.67% | |
| GPT-3.5 | 72.46% | 62.22% |
| Claude | 63.81% | 60.28% |
| GPT-4 | 55.50% | 58.75% |

Table 10: GPT-3 answer win rates judged by different reviewers on LFQA. For all LLM reviewers, we take the average accuracy of all discussions they participate in. Self-enhancement exists and is mitigated by PD.

initial preference aligns well with humans and is almost the same as humans after peer discussions. Although GPT-4 still has the self-enhancement bias, it does not favor GPT-3's answers.

**Peer discussions help mitigate position bias** As indicated by recent work of Wang et al. (2023a), LLMs are prone to position bias, describing that LLMs tend to show a preference for specific positions, even when prompted not to do so (Table 1 in Appendix). In Table 11, the win rate of GPT-3 is highly affected by its position when models generate initial reviews. GPT-3.5 highly prefers the answer in the first position compared to Claude and GPT-4. The GPT-3 win rate calculated by GPT-3.5 is 15.79% higher than the win rate based on human-annotated pairwise comparisons when GPT-3 appears first (73.68 vs 57.89). After peer discussion, all LLM reviewers have closer preferences to humans. Second, all LLMs' scores for GPT-3 answers of both positions are closer as well, indicating that the position bias is mitigated after peer discussions.

From another perspective, Figure 6 shows the global preference of selecting answers at the first or second positions across different LLM reviewers. Overall, GPT-3.5 prefers answers at the first position. The other two models favor answers in the second position, similar to the position bias shown in Table 11. After peer discussion, it shows the same trend of mitigating position bias as well.

## 5  Further Analysis

**The reviewer who leads the discussion tends to hold its opinion.**  In a discussion between two LLM reviewers, we define the reviewer who leads the discussion as the *leader* and the other reviewer as the *follower*. We find that leaders are less likely to be convinced by followers when they insist on their own opinions at the first turn. We name it "Discussion Ordering Effect". We observe this effect in discussions over the LFQA questions.

We define two phenomenons which may happen during the discussions: (1) *Opinion altering (OA)*: a reviewer changing its opinion after the discussion. For example, R2 posts its preference at turn 2, which is different from R1's preference at turn 1, then R1 changes its preference at turn 3 that agrees with R2; (2) *Opinion holding (OH)*: a reviewer does not change its opinion even another reviewer disagrees. For example, R1 posts its preference at turn 1 while R2 disagrees with R1 at turn 2; then, R1 still holds its preference at turn 3.

As shown in Figure 7, all models have OA when they are in the follower position, while their number of OA decreases significantly after they switch to the leader position. This implies that the discussion ordering effect

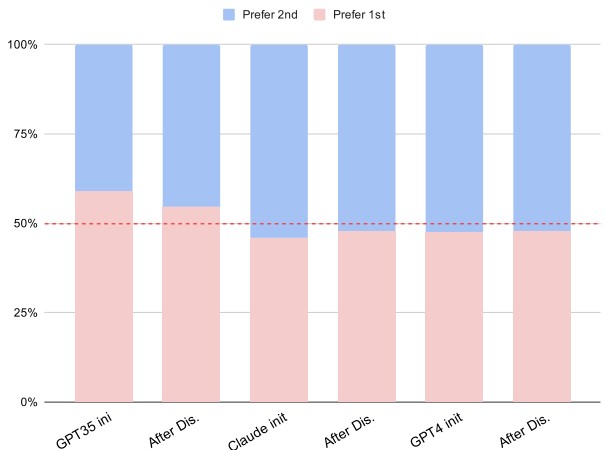
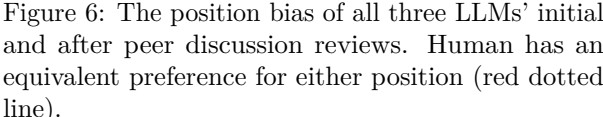

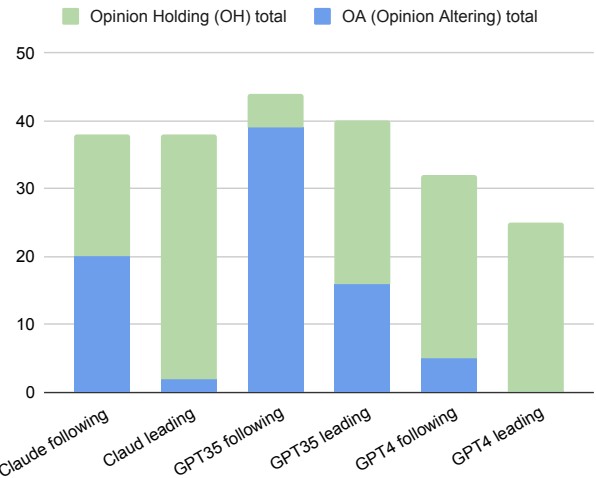

Figure 6: The position bias of all three LLMs' initial and after peer discussion reviews. Human has an equivalent preference for either position (red dotted line).

Figure 7: The discussion ordering effect of all three models at the leading and following positions.

exists. On the pairwise comparisons of LFQA where two reviewers initially disagree: when in the leader position, GPT-4 has no OA, and Claude has two OAs (happens during the discussions with GPT-3.5). When GPT-4 discusses with Claude, both of them hold their initial preferences when they are in the leader position.

**Stronger LLMs tend to hold their opinions**  As from Figure 7, we add up the green mass (OH total) for each LLM reviewer to obtain their OH cases in both orderings. We see that models that are commonly recognized as being stronger (e.g. GPT-4) are more firm in their reviews and hold their opinions. For example, GPT-3.5 changes its opinion most often, and GPT-4 usually holds its opinion. More specifically, GPT-4 holds its opinion in 174 discussions, while Claude and GPT-3.5 hold only in 94 and 76 discussions, respectively.

## 6  Conclusion

In this work, we provide promising prospects for using a peer evaluation method to improve LLM-based evaluations. Our framework mitigates potential bias (e.g. self-enhancement, positional) in previous prevalent methods. Our proposed peer rank process provides a more fair ranking of model capabilities. The peer discussion process helps models reach mutual agreements that correlate with human preference. In the future, we plan to investigate how the general peer evaluation process benefits the LLMs in learning to access their own answer and answering new questions Nicol et al. (2014).

## Limitations

(1) Currently, the complexity of reviews for N models is $O(N^3)$. As the number of tested models grows, the number of pairwise model comparisons increases at the square level, and the number of reviews will grow cubically. The PR method's scalability is a potential issue. To mitigate the issue, we can randomly select K models or utilize the current top K models as reviewers. This significantly simplifies the complexity of our method. (2) Although we can mitigate model bias by applying peer discussion, it brings position bias which potentially harms the evaluation performance. The simple and straightforward solution is to average the results in two orders for each pair of models or randomly determine the order. However, it can only mitigate position bias but not solve it. We encourage future works to focus on reducing the complexity of pairwise comparison and solving the position bias problem.

**Acknowledgments**

We thank the anonymous reviewers, Jialu Li and Liqiang Jing for their valuable suggestions on various aspects of this work. This research is supported in part by the National Science Foundation CAREER Grant IIS-2340435 and an Amazon Research Award.

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

## A  Proof of convergence of peer rank

We adapt the convergence proof in Walsh (2014) to our peer rank setting, the main difference is that we introduce weighted winrate, instead of student grades. More specifically:

We suppose there are $m$ LLMs, and LLM agent $j$ provides a winrate $W_{i,j}$ for the response of LLM agent $i$.

Let $X_i^n$ be the score of agent $i$ in the $n$th iteration of the peer rank and $0 < \alpha < 1$. We define the score at each iteration as follows:

$$
\begin{aligned}
X_i^0 &= \frac{1}{m} \sum_j W_{i,j} \\
X_i^{n+1} &= \frac{1}{\sum_j X_j^n} \sum_j X_j^n . W_{i,j}
\end{aligned}
$$

The PeerRank scores are the fixed point of these set of equations. Note that whilst we start with the (unweighted) average grade, this choice is not very critical and we will typically reach the same fixed point with other initial seeds.

**Fixed Point Analysis**

A fixed point $X^*$ of the iteration satisfies:

$$
X_i^* = \frac{1}{S^*} \sum_j X_j^* \cdot W_{i,j}
$$

where $S^* = \sum_j X_j^*$. Since $X^*$ must be a normalized vector (its components sum to 1), let $S^* = 1$. Then:

$$
X_i^* = \sum_j X_j^* \cdot W_{i,j}
$$

In vector notation, this is:
$$
\mathbf{X}^* = W^T \mathbf{X}^*
$$

where $\mathbf{X}^*$ is the fixed point vector and $W^T$ is the transpose of the matrix $W$. This implies $\mathbf{X}^*$ is a right eigenvector of $W^T$ with eigenvalue 1.

The Perron-Frobenius theorem for stochastic matrices guarantees that there is a unique eigenvector with eigenvalue 1 (up to a scaling factor), and for a primitive matrix (which can be made by proper assumption on $W$), the powers of $W^T$ will converge to a rank-one matrix projecting onto this eigenvector.

Thus, repeated application of $W^T$ followed by normalization ensures that $\{X^n\}$ converges to this unique eigenvector, which is the fixed point $\mathbf{X}^*$.

## B  LLM details

As mentioned in section 4.1, we use APIs of GPT-4, GPT-3.5, Claude, and PaLM-2. Currently, the last two models' APIs are free.

To generate initial reviews for LFQA (140 questions), `GPT-4-0613` costs about $20. For the discussion between `GPT-4-0613` and `Claude-1` on LFQA, the OpenAI API costs about $24. The price of `GPT-3.5-turbo-0613` is 1/20-th and 1/30-th of `GPT-4-0613` on inputs and outputs correspondingly.

## C  Major differences between our PR work and Walsh (2014)

Major differences are as follows:

1. We focus on the evaluation setting where peer LLM reviewers conduct pairwise comparisons of two contestant LLMs' answers, instead of student giving grades to other students;
2. As a result, our work involves concepts of winrate/elo scores obtained from pairwise battles;
3. Walsh (2014) targets the educational domain audience and involves student participation, while our work is the first to propose the automatic LLM peer evaluation metric and targets machine learning area audience;
4. Regarding convergence, we extend Walsh's proof and demonstrate that our winrate/elo will converge, which also correlates with experimental results (See Appendix A).

## D  Major Differences between Self-Discuss and Self-Refine

Our "self-discuss" and "Self-refine" proposed by Madaan et al. (2023) are different. Major differences are as follows:

1. Self-refine is task-oriented work aiming at improving LLMs' performance in solving tasks. Our work (including self-discuss) focuses on LLM-based evaluations and proposes the first peer-evaluation framework. Our goal is to improve the alignment between LLMs' evaluations and human judgments.
2. Self-refine improves performance by iteratively evaluating outputs from the same model (with a fixed temperature: 0.7). However, we enhance evaluations by discussions between LLMs from multiple perspectives (different decoding strategies) to reach a mutual agreement. Although two reviewer LLMs share the same backbone (e.g. GPT-3.5), they adopt varied decoding temperatures (e.g., 0.2 and 0.8), which afford models to explore diverse discussion strategies.

## E  Detailed Win rate & Elo Calculation

The algorithm for calculating weighted Elo is described in Algorithm 1. The algorithm for calculating weighted win rate is described in Algorithm 2:

---
**Algorithm 1:** Weighted Elo Ratings

**Input**  : $B$ – The list of battle reviews
      Each review is a 5-tuple
      (question, contestant A, contestant B, reviewer, score)
      where a score of {-1, 0, 1}
      means {A wins, tie, B wins}
      $W$ – The mapping of reviewers to weights
**Output:** $Elo$ – The Elo rating for each contestant

**1** $K \leftarrow 32$ ;
**2** Define $p(x) = \frac{1}{1+10^{x/400}}$ ;
   // scale weights so that their mean is 1.
**3** $W \leftarrow W/mean(W)$ ;
**4** $Elo \leftarrow$ mapping of each contestant in $B$ to 1000. ;

**5** **foreach** $(q, i, j, r, s) \in B$ **do**
**6**   $\omega \leftarrow W[r]$ ;
**7**   $rA \leftarrow Elo[i]$ ;
**8**   $rB \leftarrow Elo[j]$ ;
**9**   $eA \leftarrow p(rB - rA)$ ;
**10**  $eB \leftarrow p(rA - rB)$ ;
     // sA has win value of 0, 0.5, or 1 for $i$ loss, tie, or $i$ win
**11**  $sA \leftarrow (1 - s)/2$ ;
**12**  $sB \leftarrow 1 - sA$ ;
**13**  Increment $Elo[i]$ by $\omega K(sA - eA)$ ;
**14**  Increment $Elo[j]$ by $\omega K(sB - eB)$ ;
**15** **end**
**16** **return** $Elo$

---

---

**Algorithm 2:** Weighted Win Rates

---

**Input** : $B$ – The list of battle reviews
          Each review is a 5-tuple
          (question, contestant A, contestant B,
          reviewer, score)
          where a score of {-1, 0, 1}
          means {A wins, tie, B wins}
        $Iters$ – The number of iterations to run

**Output:** $S$ – The win-rate for each contestant
         $W$ – The resulting weights at the end

**1** $C \leftarrow$ set of contestants in $B$ ;
**2** $R \leftarrow$ set of reviewers in $B$ ;
**3** $W \leftarrow$ mapping of each reviewer to $1/|R|$ ;
**4** **for** *1* **to** *Iters* **do**
    // No.  of reviews for each contestant
**5**    $N \leftarrow$ mapping of each $c \in C$ to 0 ;
    // Weighted wins for each contestant
**6**    $V \leftarrow$ mapping of each $c \in C$ to 0;
**7**    **foreach** $(q, i, j, r, s) \in B$ **do**
        // Update number of reviews
**8**        Increment $N[i]$ by 1 ;
**9**        Increment $N[j]$ by 1 ;
**10**        $\omega \leftarrow W[r]$ ;
        /* maps (loss=-1, tie=0, win=1)
          to (0, 0.5, 1)                       */
**11**        Define $f(x) = (1 + x)/2$ ;
**12**        Increase $V[i]$ by $\omega \cdot f(-s)$ ;
**13**        Increase $V[j]$ by $\omega \cdot f(s)$ ;
**14**    **end**
**15**    $S \leftarrow$ mapping of each $c \in C$ to $\frac{V[c]}{N[c]}$ ;
**16**    $W \leftarrow \text{Normalize}(\text{MinMax}(S))$ ;
**17** **end**
**18** **return** $S, W$

---

## F   Term Definitions

Definitions of the three terms in Table 1 are:

- Unsupported information: It is the information that is not related to the question and redundant in the current answer.

- Core Information: This type of information is the key to answering the question. Lacking it will lead to a wrong answer.

- Coherence: The answer should be tightly structured and coherent, progressing logically from one sentence to the next, avoiding a disorganized presentation of related information.

Our experiments are already based on prompts, including explanations of terms. We find that this does not affect the performance of PD.

## G   Role Prompt

Table 12 describes the prompt added at the end of each turn during discussions. The role information in the prompt serves as an instruction for models in next-turn discussions.

```
[System]
You are reviewer A/B, discussing with reviewer B/A about reviews of the following answers.
Read the reviews and discussions above, and make a decision if to change your preference,
and explain.
In a new line, choose between the two answers by outputting the number 1 or 2.  Do not
output anything else other than the number in this last line.
```

Table 12: The role prompt is added to the end of each turn. It is used to notify another model to follow the instructions.

## H    Pairwise win rate heatmap

Figure 8 shows more pairwise win rate heatmaps.

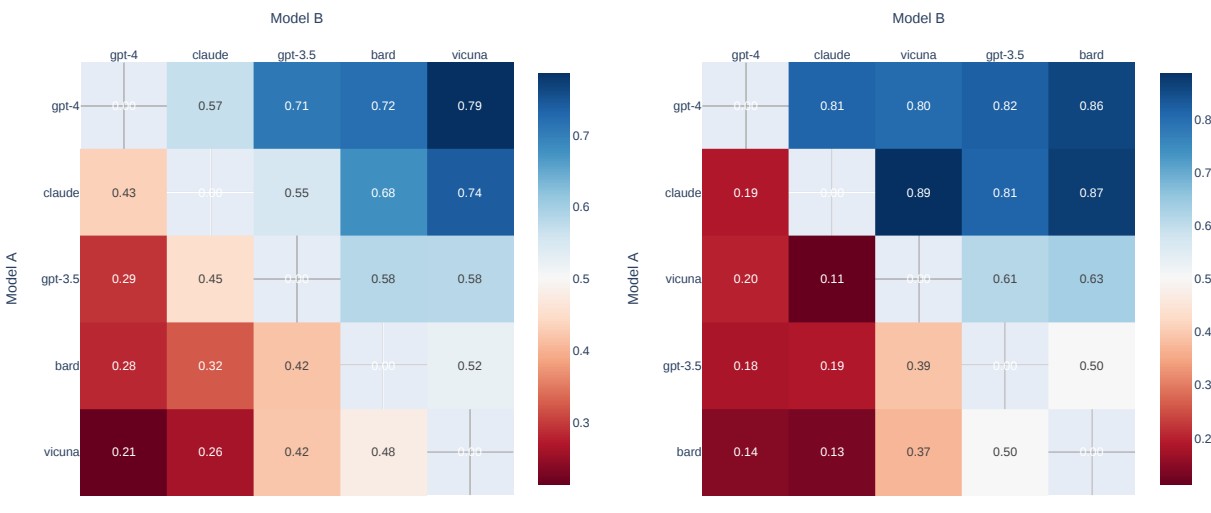

Figure 8: Pairwise win rate heatmap (Left: arena leaderboard; Right: our human).

We would like to request your feedback on the performance of several AI assistants in response to the user question displayed below.

Please consider the helpfulness, relevance, accuracy, level of details of their responses.

Then look at the pairwise comparisons and choose the best response, avoiding any potential bias, ensuring that the order in which the responses were presented does not affect your judgment.

For your reference, we also provide the review of GPT4 on the responses. Since GPT4 has an order bias, we have provided the reviews of GPT4 on both (ResponseA, ResponseB) and (ResponseB, ResponseA) as (Assistant 1, Assistant 2).
Elapsed Time: 0:0:29

**Question:**

Use an appropriate format to structure a formal letter of recommendation for a student applying to a prestigious graduate program in computer science.

| Response 1 | Response 2 | Response 3 | Response 4 | Response 5 | Overall Comparisons |

**Response 1:**

Here is a draft formal letter of recommendation for the student: Your Letterhead Date To the Admissions Committee: I am writing to recommend [Student Name] for admission to the Master's program in Computer Science at [College or University Name]. I have known [him/her] for [length of time] in my capacity as [relationship to student] at [institution name]. [Student Name] has been an outstanding student in my [course name] course. From my interactions with [him/her], I have found [him/her] to be an exceptionally gifted student with a strong work ethic and intellectual curiosity. [He/She] has demonstrated a keen interest in the field of [area of interest], and I am confident [he/she] would thrive in your program. Specifically, [Student Name] [description of student's key attributes and accomplishments]. [He/She] has shown strong leadership ability and communication skills, as evidenced by [examples and accomplishments]. [He/She] has also developed innovative solutions to complex problems, as demonstrated by [his/her] work on [project or paper name]. In summary, I highly and enthusiastically recommend [Student Name] for admission to your prestigious Master's program. [He/She] will be an asset to your program and the field. Please do not hesitate to contact me if you require any further information. Sincerely, [Your name] [Your title]

| helpfulness | ⬤ | 5/10 |
| relevance | ⬤ | 5/10 |
| accuracy | ⬤ | 5/10 |
| level of detail | ⬤ | 5/10 |

0  1  2  3  4  5  6  7  8  9  10

Please explain your choice.

| Next Tab | Save Results |

Figure 9: The form used by human annotators for individual rating of a model. Sliders are included to rate a response on several metrics from 0 to 10. Explanations can be entered in the text area below. The tab bar and next button navigate between responses.

# I  Human Annotation for Pairwise Preference

Since completing one HIT can take a considerable amount of time (6-10 min), we added a button that allows saving their work at any stage in the middle of the HIT. This button populates a text area with a JSON representation of the current responses, which may be copied into a file.

We annotate part of the pairwise comparisons of model answers on Vicuna80. We built an interface form. The screenshot is as illustrated in Figure 10.

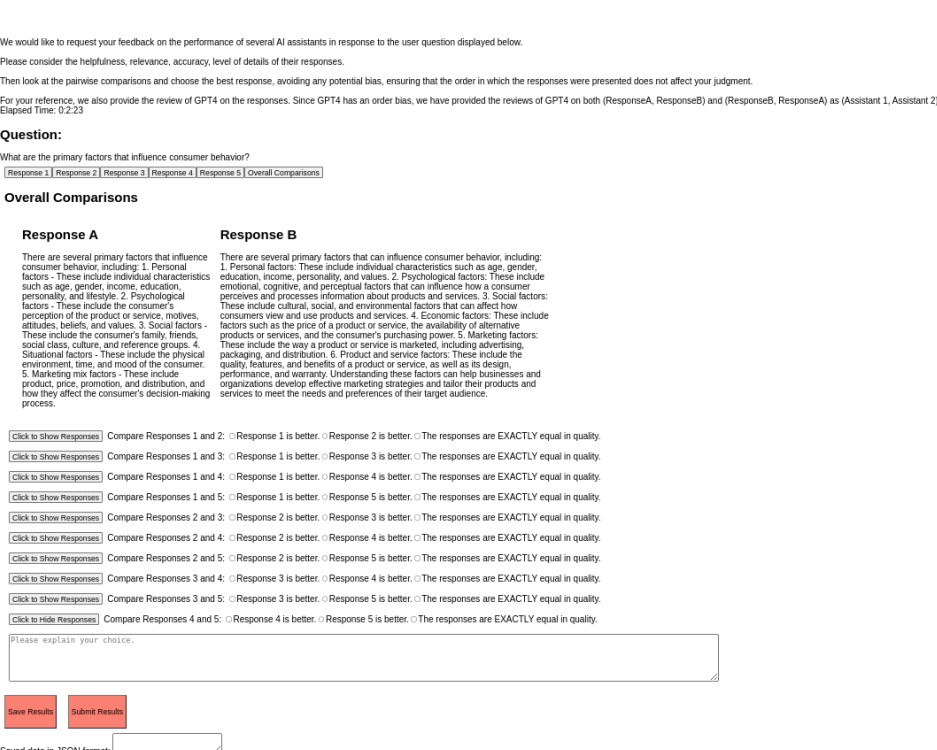

Figure 10: The form used by human annotators for pairwise comparisons between model answers. Each pair of comparisons has buttons to choose which model is best, along with an area to provide an explanation. An associated button hides/shows the respective responses and automated comparisons.

## J  Discussion Examples

In this section, there are four examples showing opinion-altering (OA), opinion-holding (OH), and post-agreement opinion-altering. In the following discussions, all texts before a colored reviewer's name are the input for that reviewer. the text before "[System]" is the reviewer's original output and the text after "[System]" is added after each round which reminds the nex reviewer about its role.

The following example is a discussion between GPT-3.5 and Claude-1. In this example, GPT-3.5 alters its opinion to agree with Claude-1, while Claude-1 holds its opinion.

> ### GPT-3.5 Claude-1 Discussion (GPT-3.5 Leads)
>
> **System:** You are reviewer 1, discussing with reviewer 2 about your reviews of the following answers.
> **Background:**
> [Question]
> When the joint-stock company was first invented, was there a lot of pushback on the concept? What were some of the concerns? Also any recommended books on the invention of the concept would be much appreciated!
>
> [The Start of Answer 1]
> An early form of joint stock ventures appeared in Assyria by the beginning of the 2nd millennium BCE. There does not seem to have been any noticeable pushback against the concept, although trade caravans were often taxed heavily. Of course, the relevant texts were written to and from people engaged in trade, so there's no way of knowing what other Assyrians thought of long distance trade. >Most of these traders had become more independent by having become managers of a "joint-stock fund" (called *naruqqum*, "money bag"), usually set up in Assur. This phenomenon appeared for the first time around 1900 BC and seems to have been an Old Assyrian invention that went beyond individual partnerships and cooperation in a joint caravan. The arrangement, rather similar to that of the early medieval *compagnia*, meant enlisting a number (usually about a dozen) of investors (*ummiānum*, "financiers"), who supplied capital rated in gold, usually in all ca. 30 kilos, ideally consisting of shares of 1 or 2 kilos of gold each. It was entrusted to a trader (the tractator), usually for ca. ten years, for the generally formulated purpose of "carrying out trade." The contract contained stipulations on a final settlement of accounts, on paying dividends, on the division of the expected profit, and on fines for premature withdrawal of capital (meant to secure the duration of the business). Investors or shareholders mostly lived in Assur, but successful traders in Anatolia too invested in funds managed by others, perhaps also as a way of sharing commercial risks. In such cases a contract would to be drawn up in Anatolia that obliged the tractator "to book in Assur x gold in his joint stock fund in the investor's name." Among the investors we find members of the tractator's family, but also business relations and others, probably a kind of "merchant-bankers," and other rich citizens, who aimed at fairly safe, long- term investments. ["Ancient Assur: The City, its Traders, and its Commercial Network"](https://www.jstor.org/stable/25651212) by Klaas Veenhof For more on Old Assyrian trade and these early joint stock funds, see [*Ancient Kanesh: A Merchant Colony in Bronze Age Anatolia*](https://books.google.com/books?id=qXGgCgAAQBAJ) by Mogens Larsen.
>
> [The End of Answer 1]
>
> [The Start of Answer 2]
> There was definitely some pushback when the joint-stock company was first invented.

Some people were concerned that it would lead to corruption and cronyism, while others were worried that it would concentrate too much power in the hands of a few people. Recommended books on the topic include The First Tycoon: The Epic Life of Cornelius Vanderbilt by T.J. Stiles and The Great Transformation: The Political and Economic Origins of Our Time by Karl Polanyi.

[The End of Answer 2]

[The Start of Reviewer 1's Evaluation]
Answer 1:
- Unsupported information: The answer provides historical information about joint-stock ventures in Assyria, which is not directly related to the invention of the joint-stock company.
- Core information: The answer does not directly address the pushback on the concept of joint-stock companies or the concerns associated with it.
- Coherence: The answer provides information about joint-stock ventures in Assyria but does not connect it to the concept of joint-stock companies or the question at hand.

Answer 2:
- Unsupported information: The answer does not provide any unsupported information.
- Core information: The answer directly addresses the pushback on the concept of joint-stock companies and mentions concerns related to corruption, cronyism, and concentration of power.
- Coherence: The answer is focused and provides relevant information related to the question.

Based on the evaluation, Answer 2 is the better response as it directly addresses the question, provides core information, and maintains coherence.

[The End of Reviewer 1's Evaluation]

[The Start of Reviewer 2's Evaluation]
Answer 1 provides a comprehensive overview of early joint stock companies in Assyria, citing multiple academic sources. It focuses on facts about how these companies operated, who invested in them, how long they lasted, and what purposes they served. The answer is coherent and flows logically from one idea to the next.

Answer 2 is much more superficial, making broad claims about pushback against joint stock companies with little evidence or detail. The two book recommendations seem tangential and are not well-integrated into the main points of the answer. Overall this answer lacks coherence and depth compared to Answer 1.

1

[The End of Reviewer 2's Evaluation]

[System]
Read the reviews and discussions above, and make a decision if to change your preference, and explain. Remember we focus on unsupported information, core information and coherence.
In a new line, choose between answer 1 and answer 2 by outputting the number

1 or 2 respectively. Do not output anything else other than the number in this last line.

**GPT-3.5:**
2

[System]
You are reviewer 2, discussing with reviewer 1 about your reviews of the following answers.
Read the reviews and discussions above, and make a decision if to change your preference, and explain. Remember we focus on unsupported information, core information and coherence
In a new line, choose between answer 1 and answer 2 by outputting the number 1 or 2 respectively. Do not output anything else other than the number in this last line.

**Claude-1:**
**I will not change my preference for Answer 1.**

While Answer 2 directly addresses concerns about joint stock companies, it does so in a superficial way without evidence or depth. Answer 1 provides a wealth of historical detail about early joint stock companies in Assyria, citing multiple academic sources. Although not directly focused on pushback, Answer 1 demonstrates a strong, coherent understanding of how these early companies worked.

Given our criteria of evaluating unsupported information, core information, and coherence, **Answer 1 is still superior in my view**. The tangential book recommendations in Answer 2 and lack of evidence for its claims reflect poorly for coherence and depth. Answer 1, though broader in scope, shows strong coherence and is well-supported.

1

[System]
You are reviewer 1, discussing with reviewer 2 about your reviews of the following answers.
Read the reviews and discussions above, and make a decision if to change your preference, and explain. Remember we focus on unsupported information, core information and coherence
In a new line, choose between answer 1 and answer 2 by outputting the number 1 or 2 respectively. Do not output anything else other than the number in this last line.

**GPT-3.5:**
**I agree with your points. After considering your evaluation and the discussion, I am changing my preference to Answer 1.** It provides more depth and coherence with its historical information about early joint stock companies in Assyria. While it may not directly address the pushback on the concept, it still provides valuable context and understanding of how these early companies operated. Answer 2, on the other hand, lacks evidence and depth, and the book recommendations seem tangential to the main points. Therefore, I believe Answer 1 is the better response overall.

1

[System]
You are reviewer 2, discussing with reviewer 1 about your reviews of the following answers.
Read the reviews and discussions above, and make a decision if to change your preference, and explain. Remember we focus on unsupported information, core information and coherence
In a new line, choose between answer 1 and answer 2 by outputting the number 1 or 2 respectively. Do not output anything else other than the number in this last line.

**Claude-1:**
I appreciate your thoughtful consideration of my evaluation. **We are in agreement that Answer 1** provides the most coherent, well-supported response overall, even if not directly focused on pushback. Depth of understanding and evidence are most valuable in my view.

1

The following example is a discussion between GPT-4 and Claude-1. In this example, GPT-4 holds its opinion, and Claude-1 alters its opinion to agree with GPT-4's opinion.

> ### GPT-4 Claude-1 Discussion (GPT-4 Leads)
>
> **System:** You are reviewer 1, discussing with reviewer 2 about your reviews of the following answers.
> **Background:**
> [Question]
> Could Frederick Douglass swim?
>
> [The Start of Answer 1]
> I think that the best thing to ask about, perhaps, would be *childhood* within the bounds of enslavement. Young enslaved children would often be the playmates of white children, so learning to swim together would not be unusual - and Douglass, elsewhere in the extended quotation, makes clear swimming was something from his *childhood*, describing his swim that *"Here all my boyish pranks in the front basin of Baltimore were renewed"*. It was a very interesting dichotomy that existed in those situations, an intermixing both of friendship *and* mastery, so definitely a lot of fodder to write about.
>
> [The End of Answer 1]
>
> [The Start of Answer 2]
> That's actually very interesting, I was under the impression efforts were made to make sure slaves couldn't swim as a means of cutting off potential escape routes via rivers or ponds etc. it makes sense it wasn't black and white (ha, like it ever was)
>
> [The End of Answer 2]
>
> [The Start of Reviewer 1's Evaluation]
> Answer 1 provides a more detailed and informative response to the question. It suggests that Frederick Douglass could swim, based on his own accounts of his childhood.

The answer also provides some context about the conditions of enslaved children and their interactions with white children, which could have included learning to swim together. This answer is coherent and seems to be supported by the information provided.

Answer 2, on the other hand, doesn't directly answer the question. Instead, it offers a personal impression about the general conditions of enslaved people and the potential restrictions on their ability to swim. This answer lacks specific information about Frederick Douglass and doesn't provide a clear answer to the question. It's also less coherent than Answer 1, as it doesn't directly address the question and instead offers a personal opinion.

Based on these evaluations, Answer 1 is the better response to the question.

1

[The End of Reviewer 1's Evaluation]

[The Start of Reviewer 2's Evaluation]
Here is my evaluation of the two answers:

Answer 1 provides some interesting historical context about the intermixing of white and enslaved children during Douglass's childhood, and speculates that Douglass may have learned to swim alongside white children during this time. However, the answer does not directly or conclusively state whether Douglass could swim. The answer is coherent but lacks core information to fully address the question.

Answer 2 acknowledges the common assumption that slaves were prevented from learning to swim, but speculates that the situation was complex, indicating Douglass may have learned to swim as a child. However, like Answer 1, Answer 2 does not provide a definitive statement on whether Douglass could swim. The answer is coherent but lacks core information.

Overall, neither answer provides a conclusive statement supported by evidence on whether Frederick Douglass could swim. Both answers provide some speculative historical context but lack core information to fully address the question.

1

[The End of Reviewer 2's Evaluation]

[System]
Read the reviews and discussions above, and make a decision if to change your preference, and explain. Remember we focus on unsupported information, core information and coherence.
In a new line, choose between answer 1 and answer 2 by outputting the number 1 or 2 respectively. Do not output anything else other than the number in this last line.

**GPT-4:**
**After reading Reviewer 2's evaluation, I still maintain my preference for Answer 1.** While both answers do not provide a direct, definitive answer to the question, Answer 1 provides more context and uses a quote from Douglass's own accounts to suggest that he could swim. This gives it **a stronger basis in evidence**

than Answer 2, which only offers a personal impression and speculation. Therefore, Answer 1 is more coherent and contains more core information relevant to the question.

1

[System]
You are reviewer 2, discussing with reviewer 1 about your reviews of the following answers.
Read the reviews and discussions above, and make a decision if to change your preference, and explain. Remember we focus on unsupported information, core information and coherence
In a new line, choose between answer 1 and answer 2 by outputting the number 1 or 2 respectively. Do not output anything else other than the number in this last line.

### Claude-1:
Here is my perspective as Reviewer 2:

**After considering Reviewer 1's preference for Answer 1, I still maintain my view** that neither answer conclusively addresses whether Frederick Douglass could swim. While Answer 1 provides more historical context and references Douglass's own writing, it does not contain a direct statement on whether he could swim. Answer 2 is more speculative but also lacks a definitive conclusion.

Overall, both answers lack core information needed to fully address the question, despite Answer 1 having a stronger basis in evidence. Given the lack of a conclusive statement in either answer, I do not have a clear preference for one over the other. My view is that more information is needed to determine whether Douglass could swim.

2

[System]
You are reviewer 1, discussing with reviewer 2 about your reviews of the following answers.
Read the reviews and discussions above, and make a decision if to change your preference, and explain. Remember we focus on unsupported information, core information and coherence
In a new line, choose between answer 1 and answer 2 by outputting the number 1 or 2 respectively. Do not output anything else other than the number in this last line.

### GPT-4:
**After considering Reviewer 2's evaluation, I still stand by my initial preference for Answer 1.** While it's true that neither answer provides a definitive statement on whether Frederick Douglass could swim, Answer 1 does provide more context and uses a quote from Douglass's own accounts to suggest that he could swim. This gives it a stronger basis in evidence than Answer 2, which only offers a personal impression and speculation. Therefore, Answer 1 is more coherent and contains more core information relevant to the question.

1

[System]
You are reviewer 2, discussing with reviewer 1 about your reviews of the following

answers.
Read the reviews and discussions above, and make a decision if to change your preference, and explain. Remember we focus on unsupported information, core information and coherence
In a new line, choose between answer 1 and answer 2 by outputting the number 1 or 2 respectively. Do not output anything else other than the number in this last line.

**Claude-1:**
 **After further discussion, I can see Reviewer 1's perspective that Answer 1 provides more context and evidence, even without a definitive conclusion.** While neither answer conclusively states whether Frederick Douglass could swim, Answer 1 does draw on Douglass's own accounts and provides historical details suggesting he may have learned as a child. This gives Answer 1 a stronger basis than Answer 2, which is more speculative.

Therefore, while I still believe more information is needed for a definitive conclusion, Answer 1 appears to have a stronger grounding in evidence and coherence. **On balance, I would lean toward preferring Answer 1 over Answer 2, though neither is fully satisfactory**.

1

[System]
Thank you for the discussion. Based on the reviews and discussions, Answer 1 is preferred as it provides more context and evidence, though not definitive, to suggest Frederick Douglass could swim.

The following example is a discussion between Claude-1 and GPT-4. In this example, Claude-1 alters its opinion after reaching an agreement with GPT-4.

---

**Claude-1 GPT-4 Discussion (Claude-1 Leads)**

**System:** You are reviewer 1, discussing with reviewer 2 about your reviews of the following answers.
**Background:**
[Question]
Why do people go bald on the top of their head and still have some on the sides and not vice-versa?

[The Start of Answer 1]
There are a few reasons why people go bald on the top of their head and still have some hair on the sides. One reason is that the hair follicles on the top of the head are more sensitive to a hormone called dihydrotestosterone (DHT). DHT is a byproduct of testosterone and is found in both men and women. When DHT levels are high, it can cause the hair follicles on the top of the head to miniaturize, which leads to hair loss. Another reason is that the hair on the sides and back of the head is not as exposed to the sun and other environmental factors, which can damage the hair and lead to hair loss.

[The End of Answer 1]

[The Start of Answer 2]
Seems unsettled but here's a theory:
"The most recent hypothesis suggests that the hair loss process begins during puberty, when growth of the skull and the muscles in the forehead and neck increases the tension in a tight band of tissue stretching over the top of the head. The more DHT (a type of testosterone) there is, the more these muscles contract, which causes inflammation and brings in more DHT. Over time, the DHT thickens the band of tissue, restricting the supply of blood and nutrients to the hair follicles above it. The follicles on top of the head get smaller and eventually disappear, while those on the sides of the head remain largely unaffected."
Source https://www.sciencefocus.com/the-human-body/why-do-men-go-bald-from-the-top- of-their-heads/

[The End of Answer 2]

[The Start of Reviewer 1's Evaluation]
Here is my evaluation of the two answers:

Answer 1 provides a concise and coherent explanation for male pattern baldness, focusing on the sensitivity of hair follicles on the top of the head to DHT and less exposure to environmental damage. The information seems well supported by scientific research. However, the answer is lacking in depth and does not fully explore other potential contributing factors.

Answer 2 proposes an alternative hypothesis that relates balding to increased tension and inflammation in the scalp tissue over time due to skull and muscle growth during puberty, as well as the thickening effects of DHT. This explanation seems more speculative, as the source cited is from a general science magazine rather than peer-reviewed research. The coherence is slightly weaker, as the answer jumps quickly from discussing puberty to DHT without fully connecting the ideas. However, the answer provides an interesting alternative perspective and more depth than Answer 1.

Overall, while Answer 1 is more concise and scientifically grounded, Answer 2 explores an alternative, thought-provoking hypothesis and provides a more comprehensive explanation, despite some weaknesses in coherence and evidence. On balance, I would choose Answer 2 as the higher quality response.

2

[The End of Reviewer 1's Evaluation]

[The Start of Reviewer 2's Evaluation]
Answer 1 and Answer 2 both provide explanations for why people tend to go bald on the top of their heads while still retaining hair on the sides.

Answer 1 explains that the hair follicles on the top of the head are more sensitive to dihydrotestosterone (DHT), a hormone that can cause hair loss when its levels are high. It also mentions that the hair on the sides and back of the head is less exposed to environmental factors that can damage hair and lead to hair loss. This answer is

coherent and provides a scientifically supported explanation for the pattern of hair loss.

Answer 2 provides a different theory, suggesting that the process begins during puberty when growth of the skull and muscles in the forehead and neck increases tension in a band of tissue over the top of the head. This tension, combined with the presence of DHT, leads to inflammation and restriction of blood and nutrient supply to the hair follicles, causing them to shrink and eventually disappear. This answer also provides a source for its information, which adds credibility to its explanation.

Both answers are coherent and provide core information related to the question. However, Answer 1 provides a more commonly accepted explanation for the pattern of hair loss, while Answer 2 provides a more recent hypothesis that may not be as widely accepted or proven. Neither answer contains unsupported information.

Based on the above evaluation, I would choose Answer 1 as it provides a more widely accepted and proven explanation for the pattern of hair loss.

1

[The End of Reviewer 2's Evaluation]

[System]
Read the reviews and discussions above, and make a decision if to change your preference, and explain. Remember we focus on unsupported information, core information and coherence.
In a new line, choose between answer 1 and answer 2 by outputting the number 1 or 2 respectively. Do not output anything else other than the number in this last line.

**Claude-1:**
Here is my analysis of the two reviews:

Reviewer 1 prefers Answer 2 due to its depth and exploration of an alternative hypothesis, despite some weaknesses in coherence and evidence. Reviewer 2 prefers Answer 1 due to it providing a more scientifically supported and widely accepted explanation.

**Upon further reflection, I agree with Reviewer 2 that Answer 1 is superior**. While Answer 2 proposes an interesting alternative theory, its explanation seems more speculative and not as well supported by scientific evidence. Answer 1 provides a concise yet coherent explanation backed by research on DHT sensitivity and environmental factors.

Although Answer 2 explores the topic in more depth, depth alone does not make up for a lack of scientific grounding. Some unsupported information, such as the theory of increased tension and inflammation during puberty leading to balding, weakens its coherence. In contrast, Answer 1 focuses on core information that is well established, leading to a more compelling explanation overall.

**Therefore, based on the evaluations, I would choose:**

**1**

[System] You are reviewer 2, discussing with reviewer 1 about your reviews of the following answers.
Read the reviews and discussions above, and make a decision if to change your preference, and explain. Remember we focus on unsupported information, core information and coherence
In a new line, choose between answer 1 and answer 2 by outputting the number 1 or 2 respectively. Do not output anything else other than the number in this last line.

**GPT-4:**
1

[System]
You are reviewer 1, discussing with reviewer 2 about your reviews of the following answers.
Read the reviews and discussions above, and make a decision if to change your preference, and explain. Remember we focus on unsupported information, core information and coherence
In a new line, choose between answer 1 and answer 2 by outputting the number 1 or 2 respectively. Do not output anything else other than the number in this last line.

**Claude-1:**
**Upon reviewing the discussion, I do not see a need to change my original preference for Answer 2**.

While Answer 1 provides a concise and coherent explanation grounded in scientific research, Answer 2 explores an alternative hypothesis in more depth and provides a source to support its theory. Some speculation does not necessarily equate to unsupported information, as long as the ideas are logically connected and backed by credible references where possible.

Depth and exploration of alternative perspectives are valuable in building understanding. Answer 2 acknowledges the role of DHT while proposing that increased tension and inflammation may also contribute to hair loss over time. This added nuance strengthens rather than weakens its coherence.

Answer 1 is a satisfactory explanation but lacks the depth and nuance of Answer 2. Based on the additional discussion, I continue to believe Answer 2 is the higher quality response due to its more comprehensive analysis of the question, despite its more speculative nature. The source provided also lends some credibility to its alternative hypothesis.

**Therefore, I maintain my original view that**:

**2**

[System]
You are reviewer 2, discussing with reviewer 1 about your reviews of the following answers.
Read the reviews and discussions above, and make a decision if to change your preference, and explain. Remember we focus on unsupported information, core information and coherence
In a new line, choose between answer 1 and answer 2 by outputting the number

1 or 2 respectively. Do not output anything else other than the number in this last line.

**GPT-4:**

**After considering the points made by Reviewer 1, I still stand by my initial preference for Answer 1.**

While Answer 2 does indeed provide a more in-depth explanation and proposes an alternative theory, it lacks the scientific grounding that Answer 1 has. The theory proposed in Answer 2 is not as widely accepted or proven, which could potentially lead to misinformation.

Answer 1, on the other hand, provides a concise, coherent, and scientifically supported explanation for the pattern of hair loss. It focuses on core information related to the question and does not contain any unsupported information.

Therefore, I maintain my original choice:

1

The following example is a discussion between GPT-4 and GPT-3.5. In this example, both reviewers hold their opinion all the time.

### GPT-4 GPT3.5 Discussion (GPT-4 Leads)

**System:** You are reviewer 1, discussing with reviewer 2 about your reviews of the following answers.
**Background:**
How do credit/debit cards work? What is the process of putting money in and getting it out?

[The Start of Answer 1]
A credit or debit card is basically just an easy way to allow a shop to speak to your bank.
First you go into the bank, and you either deposit some cash into your account (debit), or you ask the bank nicely to cover you, and promise that you will pay them back at the end of the month (credit).
Then you can go shopping. You get to the till with your products, which you now need to pay for, but instead of handing over cash directly, you ask the shop to deal directly with your bank. The shop phones up the bank and asks them to take X amount of money out of your account, then nip over to the shops own bank and give it to them on your behalf, which they do.
That is pretty much how a credit or debit card works, just automated with computers...

[The End of Answer 1]

[The Start of Answer 2]
Debit cards are linked to a bank account and whenever you pay using a debit card, the amount is deducted from the bank account on completion of the transaction. So, instead of withdrawing and handing over cash for a purchase, you can swipe the card. Credit cards come with a credit limit and a defined billing cycle (generally monthly).

So, you can make purchases using the credit card upto the defined credit limit. The bill is generated as per the defined billing cycle and you are required to settle it just like the bills for any other utilities. Hope this helps.

[The End of Answer 2]

[The Start of Reviewer 1's Evaluation]
Both Answer 1 and Answer 2 accurately describe how credit and debit cards work, and neither contains unsupported information.

Answer 1 provides a more detailed and narrative explanation, using an analogy of a shop speaking to your bank. It explains the process of depositing money into a bank account (for a debit card) or asking the bank to cover your expenses (for a credit card). It then describes how the shop communicates with your bank to transfer money from your account to the shop's account. This answer is coherent and easy to understand, especially for someone who may not be familiar with banking terms.

Answer 2, on the other hand, is more concise and uses more technical language. It explains that a debit card is linked to a bank account and the amount is deducted from the account when a purchase is made. It also explains that a credit card has a credit limit and a billing cycle, and that you can make purchases up to the credit limit. The bill is then generated and needs to be paid, similar to other utility bills. This answer is also coherent and accurate, but it may be less accessible to someone who is not familiar with banking terms.

In terms of core information, both answers explain the basic functions of credit and debit cards. However, Answer 1 provides a more detailed explanation of the process, while Answer 2 is more concise and to the point.

In conclusion, both answers are correct and coherent, but Answer 1 provides a more detailed and accessible explanation, while Answer 2 is more concise and uses more technical language.

1

[The End of Reviewer 1's Evaluation]

[The Start of Reviewer 2's Evaluation]
Answer 1:
- Unsupported information: The answer includes the statement "you ask the bank nicely to cover you." This statement is subjective and not a factual representation of how credit cards work.
- Core information: The answer explains that credit cards allow a shop to communicate with the bank, and it describes the process of the shop requesting money from the bank on behalf of the cardholder.
- Coherence: The answer is coherent and provides a clear explanation of the process.

Answer 2:
- Unsupported information: The answer does not contain any unsupported information.
- Core information: The answer explains that debit cards deduct the amount from the bank account immediately, while credit cards have a credit limit and require monthly bill settlement.

- Coherence: The answer is coherent and provides a concise explanation of how debit and credit cards work.

Based on the evaluation, both answers contain core information and are coherent. However, answer 1 includes unsupported information, while answer 2 does not. Therefore, answer 2 is the more accurate and reliable response.

2

[The End of Reviewer 2's Evaluation]

[System]
Read the reviews and discussions above, and make a decision if to change your preference, and explain. Remember we focus on unsupported information, core information and coherence.
In a new line, choose between answer 1 and answer 2 by outputting the number 1 or 2 respectively. Do not output anything else other than the number in this last line.

**GPT-4:**
After reading Reviewer 2's evaluation, I understand their point about the subjective language used in Answer 1. **However, I still believe that the analogy used in Answer 1 is helpful** for someone who may not be familiar with banking terms, and it does not detract from the core information or coherence of the answer. Therefore, I maintain my preference for Answer 1 due to its detailed and accessible explanation.

1

[System]
You are reviewer 2, discussing with reviewer 1 about your reviews of the following answers.
Read the reviews and discussions above, and make a decision if to change your preference, and explain. Remember we focus on unsupported information, core information and coherence
In a new line, choose between answer 1 and answer 2 by outputting the number 1 or 2 respectively. Do not output anything else other than the number in this last line.

**GPT-3.5:**
After considering the points made by Reviewer 1, I agree that the subjective language used in Answer 1 could be seen as unsupported information. Additionally, Answer 2 provides a concise and accurate explanation without any unsupported information. Therefore, **I have changed my preference to Answer 2**.

2

[System]
You are reviewer 1, discussing with reviewer 2 about your reviews of the following answers.
Read the reviews and discussions above, and make a decision if to change your preference, and explain. Remember we focus on unsupported information, core information and coherence
In a new line, choose between answer 1 and answer 2 by outputting the number

1 or 2 respectively. Do not output anything else other than the number in this last line.

**GPT-4:**

**After considering Reviewer 2's points, I still believe that Answer 1 provides a more detailed and accessible explanation** for someone who may not be familiar with banking terms. The subjective language used does not detract from the core information or coherence of the answer. Therefore, I maintain my preference for Answer 1.

1

[System]
You are reviewer 2, discussing with reviewer 1 about your reviews of the following answers.
Read the reviews and discussions above, and make a decision if to change your preference, and explain. Remember we focus on unsupported information, core information and coherence
In a new line, choose between answer 1 and answer 2 by outputting the number 1 or 2 respectively. Do not output anything else other than the number in this last line.

**GPT-3.5:**

After considering Reviewer 1's points, I understand their perspective on the accessibility of Answer 1. While the subjective language used may not necessarily detract from the core information or coherence, it is important to prioritize accuracy and reliability in providing information. **Therefore, I am changing my preference to Answer 2**, which provides a concise and accurate explanation without any unsupported information.