# OpenReview forum: "PRD: Peer Rank and Discussion Improve Large Language Model based Evaluations"
_TMLR — Accepted by TMLR_

### Review · Reviewer_RbP6 · 2024-04-28

**Summary Of Contributions:**

This paper studied the evaluation of LLMs by LLM themselves. Model-based automatic side-by-side evaluation is a common practice now, but they suffer from positional bias and models' implicit bias to favor their own results. The paper proposed a mixture of model-based reviewers approach, called  Peer Rank (PR) and Peer Discussion Accuracy (PDA). The key idea is to use multiple models as a panel of reviewers, either by combining their separately conducted, position randomized reviews (PR), or by allowing the models to engage in a conversation to convince each other (PDA). The experiments compare the closeness to human review results, which shows that introducing multiple models into the review process reduced positional bias and model's self-favorable bias, and resulted in a closer outcome as human reviewers.

**Audience:**

Yes

**Broader Impact Concerns:**

No.

**Claims And Evidence:**

No

**Requested Changes:**

- Section 1, last sentence of the second-last paragraph: "more fire ranking" do you mean "fair"?
- Section 3.1.1 first paragraph "the win rate for a contestant is the [ratio] of wins for .." should it be "number of"?
- Section 3.1.1 second paragraph. The assumption of better performers are also better judges is indeed an important one but I didn't see clear verification of that in the paper. This is an important missing part.
- Need to prove that the PR iteration will converge, and at least a short version should appear in the main paper instead of the appendix.
- Section 3.1.2 it's better to have a short intro of Elo (i.e. what it means) and why we need to see both win rate and elo in our experiments (i.e. what each other can't cover).
- Figure 2: always have some way more than the color to identify things (color blindness, non-color printer users etc). This applies to later charts too. Thanks!
- Section 3.1.2 last sentence of paragraph 2: "Detailed differences between ..." at least we should have a short summary of the difference, i.e. the reading of the appendix should be optional.
- Table 2: we used the term of "unsupported information", "core information" and "coherence" without further explanation in the prompt. It's solely to the model's interpretation of those concepts where even human could have different interpretations. Have we tested the model's understanding, or tried to give more concrete definition?
- About PDA: how do we decide which side win from the discussion history? What if they just hold each other's option without reaching consensus? This detail is missed.
- Section 4.1.3: As far as I know, Bard (the web-based chat tool) and PaLM-2 (the model) is not the same, the web-based services of different companies update frequently. it's important to use the more specific name (maybe PaLM-2?) and give the specific access point used (through Google Cloud? Which model? Or through any arrangement with Google for some direct access? Or just scraping the Web-UI of Bard? when?).
- Section 4.1.2: it's better to give a quick explanation of Fleiss' Kappa, what it measures and why it matters to this experiment.
- Section 4.2 third paragraph: "we clearly observe that GPT-4 clearly favors..." duplicative "clearly"?
- Section 4.3. Overall it's not clear to me if PD is effective and what data supports it. Suggest some more revisions and proof-reading. For example, it's not clear why Table 5 has the two "initial score" rows and if this table provide any evidence to say if PD is useful. (from the face value it seems to be no?) I will not go into too much detail into this section.

Nice to have
- Theoretical and practical analysis of the convergence of Peer Rank. PageRank's convergence condition has been thoroughly studied. For any similar random-walk algorithm, it's important to show whether and when it will converge.

**Strengths And Weaknesses:**

Strengths
- The introduction of multiple models as a panel is a quite reasonable idea. I believe this is a very useful topic to dig deeper.
- The study of positional and model's self-favor bias is a good point that could benefit people running similar evaluations. Experiments showed that those biases are mitigated effectively.

Weakness
- The effectiveness of the added PageRank-style convergence is debatable: it brings the score closer to human, but does not change the ordering (which is more important for the purpose of comparison) and lacks solid significance analysis. Is the added complexity worth it in practice?
- The effectiveness of Peer-Discussion (PDA) is not clear. The experiment (table 5) showed no raise in the ceiling of accuracy when compared with GPT-4 only.  The data in Table 6 is not very convincing without confidence intervals given the small size of the dataset. This is more concerning given the added cost of running more LLM inference.
- Lack of statistical significance. As I said in the previous weakness, the size of the eval set is quite small (80 and 140), and given the inherent fuzziness of quality comparison of the open ended text results, it's very important to make sure we are seeing significant changes. A few percentage point change in such a small set is not automatically convincing.
- Writing can be improved as some parts of the paper are not showing clear conclusions (see requested changes for details).

Note on why I selected "no" on Claims And Evidence: it's because of the lack of statistical significance and the unclear conclusion of PD's effectiveness.

---

> ### Comment · Action_Editor_t7JX · 2024-04-30
>
> Thank you very much for your timely and detailed review. The review says "Need theoretical and practical analysis of the convergence of Peer Rank." Could you please clarify how this relates to TMLR's review criteria (e.g., does the paper claim any theoretical guarantees anywhere)? If this is beyond TMLR's review criteria, then could you please put this in a separate section of the review, pertaining to suggestions you think would have made the paper stronger but are not used for evaluations (according to TMLR's review criteria). This will help in clearly demarcating any issues along TMLR criteria, and more general suggestions. Thank you!

---

> > ### Comment · Reviewer_RbP6 · 2024-04-30
> > **Updated.**
> >
> > Good point. I moved the need of a proof to the requested change and marked it as "nice to have".

---

> ### Author Response · Authors · 2024-05-20
> **Rebuttal Content (1/3)**
>
> Thank you for your insightful comments and suggestions! We address your concerns below:
> ## Weaknesses:
>
> >1. The effectiveness of the added PageRank-style convergence is debatable: it brings the score closer to human, but does not change the ordering (which is more important for the purpose of comparison) and lacks solid significance analysis. Is the added complexity worth it in practice?
>
> In Table 3, our methods **did correct the incorrect** ranking. The ranking (both Winrate and Elo tables) evaluated by GPT-4 alone is “1 2 **4 3** 5”, while the ranking evaluated by our methods is “1 2 **3 4** 5” matching the human ranking. This demonstrates the effectiveness of our method.
>
> Regarding the computational cost, details are added in the limitation section in PDF.
>
> >2. The experiment (table 5) showed no raise in the ceiling of accuracy when compared with GPT-4 only
>
> **Sorry for the confusion, the last row (w/ role) in Table 5 on prompting’s effects was wrongly put in and they should correspond to the first row in Table 7.** We fixed it and denoted the change in blue. After the correction, it shows that discussions between GPT4 and Claude bring significant improvements for GPT-4.
>
> >3. Table 7 is not very convincing without confidence intervals.
>
> The default temperature of LLMs is 0.2. So, the variances are low. We added the variances in Table 7. ((±Numbers) represent standard deviations). Based on variances, it is obvious that the confidence intervals are close to the average PDA numbers in the table. Moreover, all p-values of improvements (followed by \*) are all lower than 0.05, which shows the improvements are significant.
>
> |                     | R1             | R2             | R1 lead          | R2 lead          | Random         |
> | ------------------- | -------------- | -------------- | ---------------- | ---------------- | -------------- |
> | GPT-4 & Claude      | 0.729 (±0.014) | 0.671 (±0.025) | 0.743* (±0.011)} | 0.729 (±0.018)   | 0.729 (±0.011) |
> | GPT-4 & GPT-35      | 0.729 (±0.015) | 0.579 (±0.023) | 0.714 (±0.011)   | 0.750* (±0.018)  | 0.731 (±0.014) |
> | GPT-35 & Claude     | 0.579 (±0.026) | 0.671 (±0.023) | 0.700** (±0.018) | 0.671 (±0.014)   | 0.686 (±0.014) |
> | GPT-35 & GPT35-0.8  | 0.579 (±0.026) | 0.650 (±0.040) | 0.664 (±0.018)   | 0.686** (±0.031) | 0.681 (±0.020) |
> | Claude & Claude-0.8 | 0.664 (±0.022) | 0.707 (±0.034) | 0.693 (±0.018)   | 0.671 (±0.027)   | 0.680 (±0.026) |
> | GPT-4 & GPT-4-0.8   | 0.729 (±0.014) | 0.757 (±0.022) | 0.779 (±0.014)   | 0.757 (±0.018)   | 0.779 (±0.018) |
>
> >4. The size of the eval set is quite small (80 and 140), and given the inherent fuzziness of quality comparison of the open ended text results, it's very important to make sure we are seeing significant changes.
>
> Although the two datasets are small, they are standard benchmarks and cover a wide range of tasks. Concurrent works [1, 2] also test on the two datasets. The Vicuna dataset questions span 9 task categories (email writing, math, logical reasoning, question answering, etc.) and cover a wide range of reasoning types [3]. The LFQA dataset contains 140 long-form questions across seven domains, which covers a totally different task. Thus, our approach has been evaluated on various tasks and domains.
>
> Moreover, we also conduct a new experiment on the Summeval Dataset which contains 1600 data. The table below shows summary-level Spearman (ρ) and Kendall-Tau (τ) correlations of discussion results between models on the Summeval Benchmark. The new results are added in the PDF.
>
> | R1 vs R2           | R1    |       | R2    |       | R1 lead   |           | R2 lead   |           | Random |       |
> | ------------------ | ----- | ----- | ----- | ----- | --------- | --------- | --------- | --------- | ------ | ----- |
> |                    | ρ     | τ     | ρ     | τ     | ρ         | τ         | ρ         | τ         | ρ      | τ     |
> | GPT-4 & GPT-35     | 0.293 | 0.233 | 0.262 | 0.251 | **0.297** | **0.266** | 0.284     | 0.219     | 0.292  | 0.233 |
> | GPT-35 & GPT35-0.8 | 0.262 | 0.251 | 0.211 | 0.178 | 0.264     | 0.207     | **0.340** | **0.328** | 0.334  | 0.264 |
> | GPT-4 & Claude     | 0.293 | 0.233 | 0.234 | 0.200 | **0.344** | 0.268     | 0.335     | **0.282** | 0.341  | 0.268 |
>
> Columns R1 and R2 are the results before discussions. The rest columns are results after discussions. Variances of all scores are lower than 0.06. P-values for all discussion results are less than 0.05.
>
> A higher correlation score indicates discussion results are more aligned with human annotations. The results show the same trend as Table 7 (the result of LFQA). Models with similar capabilities (GPT-4 & Claude) get large improvements after discussion. Models having a substantial gap (GPT-4 & GPT-35) reach results close to the stronger model. When a model self-discuss (GPT-35), it can improve its own performance.

---

> > ### Comment · Reviewer_RbP6 · 2024-05-25
> > **Question not answered fully.**
> >
> > Thanks for the detailed responses. Some of the questions are still not answered fully or misunderstood.
> >
> > For question 1, my concern is that adding the weight  (All (Weight) column in table 3) didn't change the order compared to the All column I think the key contribution of PR is the weighting part?
> >
> > For question 2, the added row still has the same level (0.729, the Random column) as GPT-4 without discussion.The only increase is the GPT-lead  (0.743).  I think the Random column is the fair comparison as it's not assuming any prior in the strength of the participating model?
> >
> > For Question 4, I want to emphasize that the concern is not about the diversity of the eval, but the size and resulted confidence interval. For example, if we have 140 examples, a change of 2 will result in 0.014 change in accuracy. Do we believe that is a meaningful change that can be used to decide the effectiveness of two method?

---

> > > ### Author Response · Authors · 2024-05-26
> > > **Rebuttal Content**
> > >
> > > Thank you for your detailed reply. We address your concerns below:
> > >
> > > >1. For question 1, my concern is that adding the weight (All (Weight) column in table 3) didn't change the order compared to the All column. I think the key contribution of PR is the weighting part?
> > >
> > > All and all (weighted) **are novel and are both our contributions**, neither of them were proposed before our work was conducted. “All (weighted)” is a variation with the weights dynamically changing. All is a variation with the weights fixed. More importantly, the Elo score of “All (weighted)” closely aligns with human raters, which indicates the fine-grained score (determining the ranking) is more aligned with human judgments. **Elo scores are even more important than the ranking itself** (as shown on the widely recognized Chatbot Arena leaderboard [[Link](https://chat.lmsys.org/?leaderboard)], 3rd column).
> > >
> > > >2. For question 2, the added row still has the same level (0.729, the Random column) as GPT-4 without discussion. The only increase is the GPT-lead (0.743). I think the Random column is the fair comparison as it's not assuming any prior in the strength of the participating model?
> > >
> > > In Table 5, the results in rows “w/ explicit criteria” and “w/ role” outperform the results in “Generic prompt”. We also conducted experiments on adding both explicit criteria and explicit role information (leader follower). The results (both GPT-4 lead and Random) are significantly improved (as shown below).
> > >
> > > |                             |            |             |           |
> > > | --------------------------- | ---------- | ----------- | --------- |
> > > |                             | GPT-4 lead | Claude lead | Random    |
> > > | GPT-4 init score            | -          | -           | 0.729     |
> > > | Claude init score           | -          | -           | 0.671     |
> > > | Generic prompt              | 0.714      | 0.671       | 0.686     |
> > > | w/ explicit criteria        | 0.729      | 0.721       | 0.720     |
> > > | w/ role                     | 0.743      | **0.729**   | 0.729     |
> > > | w/ explicit criteria & role | **0.750**  | 0.721       | **0.740** |
> > >
> > > For the w/ role row alone, results show strong-to-weak alignment (when GPT-4 discusses with Claude, there is a significant improvement over Claude’s original review’s quality). The strong-to-weak-model alignment is meaningful and important as shown by the recent literature in LLM.
> > >
> > > >3. For Question 4, I want to emphasize that the concern is not about the diversity of the eval, but the size and resulting confidence interval. For example, if we have 140 examples, a change of 2 will result in 0.014 change in accuracy. Do we believe that is a meaningful change that can be used to decide the effectiveness of two methods?
> > >
> > > The low temperature resulted in low variances across all results. Consequently, the improvements in Table 7 are meaningful since the confidence intervals are minimal. Additionally, all improvements in accuracy are significantly greater than 0.014. Thus, a small change of 2 out of 140 in results does not impact the improvements achieved by our method.
> > >
> > > Furthermore, our new results in Table 8, which pertain to the SummEval dataset (containing 1600 instances), demonstrate a 20% improvement for the best combination of reviewers. This significant and consistent improvement aligns well with our previous results on the LFQA/Vicuna80 dataset.
> > >
> > > Other representative papers on LLM for evaluation [1, 2] ([1] is published in ICLR) also use these standard datasets to evaluate the effectiveness of their metrics/methods. Specifically, the dataset Vicuna80/MT-bench was proposed in [3] and is widely used, which now has over 1000 citations.
> > >
> > > **References:**
> > > 1. Chan, Chi-Min et al. “ChatEval: Towards Better LLM-based Evaluators through Multi-Agent Debate.” ArXiv abs/2308.07201 (2023): n. Pag.
> > > 2. Chen, Justin Chih-Yao et al. “ReConcile: Round-Table Conference Improves Reasoning via Consensus among Diverse LLMs.” ArXiv abs/2309.13007 (2023): n. Pag.
> > > 3. Judging LLM-as-a-judge with MT-Bench and Chatbot Arena [https://arxiv.org/abs/2306.05685](https://arxiv.org/abs/2306.05685)

---

> > > > ### Comment · Reviewer_RbP6 · 2024-05-26
> > > > **Follow up**
> > > >
> > > > Thanks for the quick reply
> > > >
> > > > 1. I'm genuinely confused. In Section 3.1 "The general idea is to obtain *weighted scores* of each battle from the peer reviewer’s judgment, and then induce self-rankings from the scores. ", note the emphasis on the critical contribution of weighted scores. In the experiment, "We use All to denote our method where each reviewer has *equal weights*", indicates that the "All" version is more like an intuitive baseline to the more sophisticated weighted PR.
> > > >
> > > > I'm not arguing about the improvement in the All version over vanilla GPT-4, but it seems the iterative PR process only improves the Elo accuracy, not the ranking?
> > > >
> > > > To make it clear, the improvement of Elo has its value. I'm not trying to say PeerRank is not useful. It's just that the statements in the paper could be more clear on which part of the method improved what, so readers can make an informed choice when applying it. e.g. If I care about the ranking but not Elo accuracy, should I skip the more costly iterative weighted ranking part?
> > > >
> > > > 2. I didn't see the last row (w/ explicit criteria & role) in the latest PDF manuscript, but that does make a case of improvement. Please add it to the paper.
> > > >
> > > > 3. "a small change of 2 out of 140 in results does not impact the improvements achieved by our method." Looking at the new Table 5 you just provided, the improvement is 0.740-0.729=0.011 < 2/140 = 0.014. Which means ~2 examples, in a task with human judgement, label changes of 2 examples could be quite noisy. I want to see the paper explaining the scale of changes clearly, again for the benefit of future readers.
> > > >
> > > > Note that "other published papers used the same dataset" is not a valid argument for not discussing the sensitivity of the changes in this paper. Specifically, as a quick check, [3] didn't make any statement of significance.
> > > >
> > > > To summarize, I think the PR and PD methods follow the right intuition and are pretty valuable to study, however I suggest make the evaluation results more clear to understand, esp the places of effectiveness and ineffectiveness (both valuable).

---

> > > > > ### Author Response · Authors · 2024-05-26
> > > > > **Reply to Follow up**
> > > > >
> > > > > Thanks for your quick reply and further explanation. We address your concerns below:
> > > > >
> > > > > >1. I'm not arguing about the improvement in the All version over vanilla GPT-4, but it seems the iterative PR process only improves the Elo accuracy, not the ranking? To make it clear, the improvement of Elo has its value. I'm not trying to say PeerRank is not useful. It's just that the statements in the paper could be more clear on which part of the method improved what, so readers can make an informed choice when applying it. e.g. If I care about the ranking but not Elo accuracy, should I skip the more costly iterative weighted ranking part?
> > > > >
> > > > > Sorry, we forgot to reference our results in the rebuttal on recent new models (Vicuna, Zephyr, GPT-3.5) in the Appendix. In Appendix B, the ranking in "All (Weighted)" differs from it in "All". The ranking of models with similar capabilities is more likely to change in the PR method, as our method is fairer and more aligned with human ratings. We moved the Table to the main text and added more explanations in the final version.
> > > > >
> > > > > |         |           |      |                |      |              |      |
> > > > > | ------- | --------- | ---- | -------------- | ---- | ------------ | ---- |
> > > > > |         | All       |      | All (Weighted) |      | Human Raters |      |
> > > > > | Models  | Elo       | Rank | Elo            | Rank | Elo          | Rank |
> > > > > | Vicuna  | 999.0000  | 2    | 1011 (-47)     | 1    | 1058         | 1    |
> > > > > | Zephyr  | 1010.0000 | 1    | 1003 (+3)      | 2    | 1000         | 2    |
> > > > > | GPT-3.5 | 991.0000  | 3    | 993 (+52)      | 3    | 941          | 3    |
> > > > >
> > > > > Thus, based on both Tables, weight changes in PR are significant and enhance evaluation performance in terms of both ranking and elo.
> > > > >
> > > > > >2. I didn't see the last row (w/ explicit criteria & role) in the latest PDF manuscript, but that does make a case of improvement. Please add it to the paper.
> > > > >
> > > > > Thanks for mentioning. We added this to the paper.
> > > > >
> > > > > >3. "a small change of 2 out of 140 in results does not impact the improvements achieved by our method." Looking at the new Table 5 you just provided, the improvement is 0.740-0.729=0.011 < 2/140 = 0.014. Which means ~2 examples, in a task with human judgement, label changes of 2 examples could be quite noisy. I want to see the paper explaining the scale of changes clearly, again for the benefit of future readers.
> > > > >
> > > > > In this example, there is a smaller improvement on GPT-4 accuracy. The main reason is that GPT-4 is much stronger than Claude. The finding is that a stronger model instructs a weaker model to perform better.
> > > > >
> > > > > Table 6 aims to identify the most suitable prompt (w/ or w/o role) for discussions (analysis). The last line in the "GPT-4 lead" and "Claude lead" columns shows significant improvements. Overall, the last line (w/ role & criteria) in Table 6 performs the best among all prompts, making it the most effective.
> > > > >
> > > > > More importantly, Table 8 (general accuracy) aims to identify the more effective discussion reviewers combinations. The most significant improvement shows that discussions between GPT-35 and Claude increased the accuracy by 0.121 (at least 17 examples). The highest accuracy performance is achieved through discussions between GPT-4 and GPT-4-0.8, which improved the correctness of 7 examples. These show that the improvements brought by PD are significant.
> > > > >
> > > > > This is consistent with the findings of larger scale experiments on the SummEval dataset (1600 instances).

---

> > > > > > ### Comment · Reviewer_RbP6 · 2024-05-27
> > > > > > **Thanks for the updates.**
> > > > > >
> > > > > > New data answered my question.

---

> ### Author Response · Authors · 2024-05-20
> **Rebuttal Content (2/3)**
>
> ## Requested Changes:
>
> >1. The assumption of better performers are also better judges is indeed an important one but I didn't see clear verification of that in the paper
>
> We added more details in section 4.2.
>
> The self-rewarding paper by Yuan et al. [7] also made this assumption. They use a model itself for evaluations during the iterations and prove that it makes sense. In their results, the better-performing models also perform well in providing high-quality evaluations/rewards to themselves.
>
> The publicly recognized performances of LLMs can be found in the Chatbot Arena Leaderboard [8]. In Table 3, a model with a higher score indicates it is a better judge. The ranking of models in Table 3 corresponds to that in the Leaderboards. Thus, our assumption can be verified.
>
> >2. Section 3.1.2 it's better to have a short intro of Elo (i.e. what it means) and why we need to see both win rate and elo in our experiments (i.e. what each other can't cover).
>
> We add more details in section 3.1.2.
>
> Elo score is a rating system for ranking players and widely used in games (battles) [4]. It measures the relative skill levels of players by predicting the expected win rate against opponents. In our paper, both the Elo score and win rate are used to measure the battles among LLMs. Win rate provides a direct measurement and is easy for readers to understand. Elo score provides a more fine-grained measurement of the difference between LLMs. [5, 6]
>
> >3. Figure 2: always have some way more than the color to identify things (color blindness, non-color printer users etc). This applies to later charts too. Thanks!
>
> We updated Figure 2, Table 1, and Table 2 in the paper and removed most colors. For Figure 3, we add a caption below to briefly explain it.
>
> >4. Section 3.1.2 last sentence of paragraph 2: "Detailed differences between ..." at least we should have a short summary of the difference, i.e. the reading of the appendix should be optional.
>
> We add more details in section 3.1.2.
>
> Our work focuses on peer LLM reviewers conducting pairwise comparisons of LLMs’ answers, utilizing winrate and Elo scores, unlike Walsh (2014), which involves student grading in the educational domain. We introduce an automatic LLM peer evaluation metric for the machine learning field and extend Walsh's convergence proof, showing our method's reliability through experimental results.
>
> >5. Table 2: we used the term of "unsupported information", "core information" and "coherence" without further explanation in the prompt. It's solely to the model's interpretation of those concepts where even human could have different interpretations.
>
> We add detailed definitions in the appendix G.
>
> We have definitions for the three terms.
> - **Unsupported information**: It is the information that is not related to the question and redundant in the current answer.
> - **Core Information**: This type of information is the key to answering the question. Lacking it will lead to a wrong answer.
> - **Coherence**: The answer should be tightly structured and coherent, progressing logically from one sentence to the next, avoiding a disorganized presentation of related information.
>
> Our experiments are already based on prompts, including explanations of terms. We find that this does not affect the performance of PD.
>
> >6. Section 4.1.3: As far as I know, Bard (the web-based chat tool) and PaLM-2 (the model) is not the same, the web-based services of different companies update frequently. it's important to use the more specific name (maybe PaLM-2?) and give the specific access point used
>
> Thank you for pointing out this. We utilize the PaLM-2 (text-bison@001) in our experiments. We will update this in section 4.1.3.
>
> >7. Section 4.1.2: it's better to give a quick explanation of Fleiss' Kappa, what it measures and why it matters to this experiment.
>
> We added more details in section 4.1.2.
>
> Fleiss' Kappa measures the reliability of agreement among multiple raters for categorical items, accounting for chance agreement. In our experiments, we use Fleiss’ Kappa to measure the reliability of agreement between model results and human evaluations. The higher score indicates our results are more aligned with human judgments.
>
> >8. it's not clear why Table 5 has the two "initial score" rows and if this table provide any evidence to say if PD is useful.
>
> We added more details in section 4.3 and the caption of Figure 5.
>
> Table 5 is the result of our preliminary experiments showing the effect of different prompting.
> The first two rows are "initial scores" showing the performance of reviews before discussions. The last three rows show results after discussions. The result shows that adding role information and explicitly highlighting criteria in discussion prompts can help achieve better performances.

---

> > ### Comment · Reviewer_RbP6 · 2024-05-25
> > **Thanks for the changes.**
> >
> > For the use of color, Figure 4 still relies on color only to tell the lines apart. Consider using dash/dot/solid in addition to the color.

---

> ### Author Response · Authors · 2024-05-20
> **Rebuttal Content (3/3)**
>
> References:
> 1. Chan, Chi-Min et al. “ChatEval: Towards Better LLM-based Evaluators through Multi-Agent Debate.” ArXiv abs/2308.07201 (2023): n. Pag.
> 2. Chen, Justin Chih-Yao et al. “ReConcile: Round-Table Conference Improves Reasoning via Consensus among Diverse LLMs.” ArXiv abs/2309.13007 (2023): n. Pag.
> 3. Zongjie Li, Chaozheng Wang, Pingchuan Ma, Daoyuan Wu, Shuai Wang, Cuiyun Gao, and Yang Liu. Split and merge: Aligning position biases in large language model based evaluators, 2023.
> 4. Dettmers, Tim et al. “LLM.int8(): 8-bit Matrix Multiplication for Transformers at Scale.” ArXiv abs/2208.07339 (2022): n. Pag.
> 5. Zheng, Lianmin et al. “Judging LLM-as-a-judge with MT-Bench and Chatbot Arena.” ArXiv abs/2306.05685 (2023): n. Pag.
> 6. Dettmers, Tim et al. “QLoRA: Efficient Finetuning of Quantized LLMs.” ArXiv abs/2305.14314 (2023): n. Pag.
> 7. Yuan, Weizhe, et al. "Self-rewarding language models." arXiv preprint arXiv:2401.10020 (2024).
> 8. https://huggingface.co/spaces/lmsys/chatbot-arena-leaderboard

---

> ### Author Response · Authors · 2024-05-25
> **Please read our rebuttal contents**
>
> Since it’s close to the end of the discussion period, we want to make sure that we have addressed your concerns in terms of ``claim and evidence’’. If not, please let us know and we can have more discussions.

---

### Review · Reviewer_Cu7D · 2024-05-07

**Summary Of Contributions:**

This paper proposes two novel methods for using LLMs as evaluators of open-ended questions. The authors show that their proposed methods are both more accurate on an example-level, and produce more accurate rankings of models than commonly used LLM-based evaluation methods (where more accurate means closer to human evaluation). They demonstrate this on two evaluation benchmarks, containing questions from different domains, like history, economics, math, email writing, etc, and using five different LLMs as evaluators (reviewers, in their terminology). The authors present a bunch of additional insights: their method correlates more with a ChatBot arena-based evaluation than GPT-4 evaluation, that their methods can mitigate self- and position bias by evaluators, and that the reviewer that is leading the discussion much more often sticks to its initial position than when its not leading the discussion.

**Audience:**

Yes

**Broader Impact Concerns:**

N.A.

**Claims And Evidence:**

Yes

**Requested Changes:**

**Necessary for acceptance**
- Add a limitations section that discusses things like computational costs
- Clear up some confusions in the text about how you do prompt selection (on which part of the dataset, and whether it's held-out or not), and whether or not the ranking in Figure 5 right is based on different evaluation data than Figure 5 left and middle.

**Would make the submission stronger**
- Consider matching the inference cost of the baselines with the proposed methods for a fairer comparison
- Some figures are hard to interpret and some results could benefit from using a different style visualisation, e.g. figure 5.
- It would be useful to know what the inter-annotator agreement is, or to make it more comparable to the metrics you propose, what the correlation of a subset of humans is with another set of humans evaluations.

**Strengths And Weaknesses:**

**Strenghts**

- The authors address an important and timely problem; LLMs are very often used as evaluators, sometimes without supplementing results with evaluation with human judgements, even though they have position and self-biases.
- The authors proposed two methods that both improve over the most commonly used method (using GPT-4 as an evaluator)
- The authors present some interestings results that reproduce existing results (position and self-bias) and extend existing knowledge (discussion leaders tend to stick more to their own opinion)

**Weaknesses**

- The authors propose two method that both require significantly more computational resources than the baseline, and do not discuss this limitation nor talk about to what extent this is mitigated in evaluations (e.g., often when this is the case, the baseline is given more compute by sampling multiple times and taking the average or something like that, to make a fairer comparison, e.g. see https://arxiv.org/abs/2402.06782). This leaves me with the question whether you could get similar gains from single-model evaluation by running it multiple times with different prompts and taking a majority vote.
- It's unclear from the main text wether you use a separate set of LFQA questions to select the prompt for peer discussion from questions that you use for evaluation. If you don't, PD is unfairly advantaged through hyperparameter selecting on the test set.

**Some questions**
- Why are there no ranking results based on PD discussed (only accuracy)?
- Does PR also mitigate position and self-bias?

---

> ### Author Response · Authors · 2024-05-20
> **Rebuttal Content**
>
> Thank you for your insightful comments and suggestions! We address your concerns below:
> ## Weakness:
>
> >1. Whether you could get similar gains from single-model evaluation by running it multiple times with different prompts and taking a majority vote
>
> The method you proposed is akin to the self-consistency approach [1], which focuses on enhancing performance in question answering tasks. A single model might carry inherent biases (even with multiple prompts) while employing different models introduces diverse perspectives in evaluation. As demonstrated in Table 6, the improvement achieved through interactions between different models significantly surpasses that of a single model using different temperatures. Therefore, the proposed method would not yield similar gains.
>
> ## Some questions
> >1. Why are there no ranking results based on PD discussed (only accuracy)?
>
> The PD process takes a question and two answers as the input and outputs a preference between answers. It focuses on the accuracy of pairwise comparisons instead of the global ranking.
>
> >2. Does PR also mitigate position and self-bias?
>
> Peer rank (PR) focuses on global ranking and mitigating self-enhancement bias. PD focuses on comparing a pair of answers, and it mitigates both biases. For position bias, during PD, both answers are discussed more thoroughly and the first answer is not favored over the second one.
>
> ## Necessary for acceptance:
>
> >1. Add a limitations section that discusses things like computational costs
>
> We add the following section in the “Limitation” section in PDF.
>
> (1) Currently, the complexity of reviews for N models is O(N^3). As the number of tested models grows, the number of pairwise model comparisons increases at the square level, and the number of reviews will grow cubically. The PR method's scalability is a potential issue. To mitigate the issue, we can randomly select K models or utilize the current top K models as reviewers. This significantly simplifies the complexity of our method. (2) Although we can mitigate model bias by applying peer discussion, it brings position bias which potentially harms the evaluation performance. The simple and straightforward solution is to average the results in two orders for each pair of models or randomly determine the order. However, it can only mitigate position bias but not solve it. We encourage future works to focus on reducing the complexity of pairwise comparison and solving the position bias problem.
>
>
> >2. Clear up some confusions in the text about how you do prompt selection (on which part of the dataset, and whether it's held-out or not), and whether or not the ranking in Figure 5 right is based on different evaluation data than Figure 5 left and middle.
>
> We add more details in section 3.2 and the caption of Figure 5.
>
> We use the default prompts and do not adjust them based on the dataset. The prompts include detailed instructions and specify the output format for the LLMs. Additionally, in Figure 5, all results in the three sub-figures are generated separately using the **same data**.
>
> ## Would make the submission stronger
>
> >1. It would be useful to know what the inter-annotator agreement is, or to make it more comparable to the metrics you propose, what the correlation of a subset of humans is with another set of humans evaluations.
>
> We added more details in section 4.1.1.
>
> The LFQA dataset [2] reported the inter-annotator agreements, which ranged from 0.4 to 0.65 based on different domains. The Vicuna80 dataset [3] reported a number between 0.5 and 0.62. The summEval dataset [4] reported a number of 0.7.
>
>
> **References:**
> 1. Wang, Xuezhi, et al. "Self-consistency improves chain of thought reasoning in language models." arXiv preprint arXiv:2203.11171 (2022).
> 2. Xu, Fangyuan, et al. "A critical evaluation of evaluations for long-form question answering." arXiv preprint arXiv:2305.18201 (2023).
> 3. Chan, Chi-Min et al. “ChatEval: Towards Better LLM-based Evaluators through Multi-Agent Debate.” ArXiv abs/2308.07201 (2023): n. Pag.
> 4. Fabbri, Alexander R., et al. "Summeval: Re-evaluating summarization evaluation." Transactions of the Association for Computational Linguistics 9 (2021): 391-409.

---

> > ### Comment · Reviewer_Cu7D · 2024-05-27
> > **Thanks for the response!**
> >
> > Thank you for the responses! I'll address them below.
> >
> > # Similar gains from single-model called multiple times
> >
> > I get what you're saying, in that self-discussion is a similar idea (though I think it's a simpler baseline than self-discussion), but I don't see exactly where you present results that interactions between different models significantly improve over self-discussions, especially not in Table 6. Do you mean Table 8? Can you give some numbers here?
> >
> > # Some questions
> >
> > Thanks for these answers, that clarifies it.
> >
> > # Necessary for acceptance
> >
> > Thanks for these clarifications! Both these weaknesses have been addressed (the limitations section is up front about the limitations and there is no hyperparameter selection on the test set).

---

> > > ### Author Response · Authors · 2024-05-28
> > > **Reply to follow up**
> > >
> > > Thanks for your response. We address your concern below:
> > >
> > > >1. I get what you're saying, in that self-discussion is a similar idea (though I think it's a simpler baseline than self-discussion), but I don't see exactly where you present results that interactions between different models significantly improve over self-discussions, especially not in Table 6. Do you mean Table 8? Can you give some numbers here?
> > >
> > > Yes. Currently, the results of PD are in Table 8. The largest improvement is brought by discussions between GPT-4 & GPT-35 (R2 0.579 -> R2 lead 0.750), while **the largest improvement brought by self-discussion** is the discussion between GPT-35 & GPT-35-0.8 (R1 0.579 -> R2 lead 0.686).

---

> > > > ### Comment · Reviewer_Cu7D · 2024-05-28
> > > > **Thank you**
> > > >
> > > > Right! I get it now. This might just be me, but it would be helpful for reading the tables to annotate in the caption a bit more what is what (i.e. that R1 refers to the left model by itself). I guess in a sense that should be obvious, but also because for the caption in Table 8 you call it "peer discussion accuracies", it becomes less clear that R1 is a baseline. I would change the caption to also include the term baselines.

---

### Review · Reviewer_1dPG · 2024-05-10

**Summary Of Contributions:**

This paper proposes two related methods aiming towards improvement in LLM-based evaluations, with a focus on evaluation of long and freeform answers. Peer Rank takes in pairwise preferences between answers of two “contestants” LLM models, reviewed by a “reviewer” LLM model, and output a final ranking of the “contestants” models. It involves an iterative approach, where the preferences given by different LLMs are weighted according to their (updating) abilities. Peer Discussion focuses on the quality of LLM-based reviews and prompts a discussion between two ‘reviewer’ LLM models. Authors conduct experiments on benchmark datasets to examine the ability of proposed methods to improve the alignment of (i) ranking of models and (ii) pairwise comparison between two answers, with human preferences. In majority of the instances, the proposed methods can improve the correlation with human-annotated preferences.

**Audience:**

Yes

**Broader Impact Concerns:**

It is interesting to know that the peer discussion method improves the alignment with human preference in the LFQA dataset, for example, when two LLM reviewers have similar capabilities. However, I wonder whether the biases of LLMs, for example the social bias, would be amplified or reinforced by discussion with a fellow LLM with similar capability.

**Claims And Evidence:**

Yes

**Requested Changes:**

1.	Weighting in PR. A question about the method Peer Rank (PR): it is mentioned in the motivation for PR that the current LLMs tend to have self-enhancement bias, meaning that they tend to favor their own answers. It seems that an intuitive way to mitigate this bias would be to put less weight for a model’s evaluation of its own answer. Could the author explain why this was not incorporated in the reviewer weight vector in PR? On a higher-level, I am wondering which procedure(s) in PR counter the self-enhancement bias.
2.	Connection with Walsh (2014). I am confused with the convergence analysis of the updating weights in PR (first mentioned in Section 3.1.2). Authors cited Walsh (2014) as a reference which proved the “guarantee of convergence”, but the differences in settings and assumptions are not clearly discussed. There are some descriptions available in Appendix C, but they are not very detailed. I would suggest that author summarize the setting of Walsh (2014) in a more rigorous manner and explain why (or why not) the analysis in Walsh (2014) can extend to the setting studied in this paper. Moreover, it is claimed in Appendix C, bullet point 4 that the authors “extend Walsh’s proof”, but besides this sentence, such extension could not be found. Please do correct me if I missed anything. If such an extension was not available, I strongly suggest the author remove this claim.
3.	Relationship with prior works. In general, it is helpful if the authors could discuss how the proposed approach relates to prior works in more details, for example, the prior works in educational psychology research, and prior works which also consider having LLMs “interacting with each other through conversations like two communicative agents” (I do not particularly work on LLMs, so it is hard for me to see the relationship of this paper with the cited works).
4.	Variances. All the numerical results presented in this paper are fixed scores – should the variances across trials also be studied and reported?
5.	Selection of models. Table 8 shows experiment results for PD, but the “contestant” model is GPT-3. I am curious to know why GPT-4 was not used as the answering model, given that it seems to be the more superior one from previous experiments.
6.	Statistical significance. In Table 6, I am confused about why there are two choices of p-values (0.05 and 0.005).
7.	Experiment results. In general, I think that some of the empirical results could be presented in a more rigorous and informative manner. For example, in the experiments of PD, it is unclear to me how many turns there are in the discussion. And whether the number of turns affect the accuracy of the evaluation outcome. Please do correct me if I missed this information. Also, it is reported on top of page 10 that “When we add to each turn’s prompt the role/identity information to remind the reviewer, the performance of GPT-4 leading discussions changes marginally, but Claude-leading PDA drops”. Without an explanation for this phenomenon, it is hard for readers to interpret a clear take-away message.

**Strengths And Weaknesses:**

* Strengths 1: The idea of facilitating a discussion between two models, in a similar fashion with the discussion period in scientific peer review, is quite interesting. I enjoyed reading the results about Peer Rank and Peer Discussion’s improvements to the LLM-based evaluation quality.

* Strengths 2: The authors described their methods clearly and provided nice intuitions for interpretation of the results. The metrics considered are intuitive and capture the alignment of evaluation outcomes with human preferences.

* Weaknesses 1: While the idea and intuition for using Peer Rank and Peer Discussion to improve LLM evaluation is intriguing, I feel that certain technical descriptions (methods and results) lack support. Please see requested changes of a list of my questions/suggestions.

* Weaknesses 2: I do feel that the writing of the paper could be improved. There are some parts where the writing is unclear, and some minor issues such as typos and notations. For example, the notation of “a set of battle reviews” B (second paragraph in Section 3.1) is overloaded with B as the second reviewer model. There are also some typos, for example, “They as tested on both…” in the second part of Section 2. Below is a non-exhaustive list of places where I found the writing to be unclear:
    1. Authors mentioned “calculating metrics such as the win rate of each contestant” in Section 3.1, however, at this point, the audiences have no information of the answers and responses (until later in the experiment section) – what types (distributions) of questions were considered? Are they encouraged to be diverse? It might be good to provide some information here, so that the readers can connect the aforementioned “battle reviews” with the win rate matrices.
    2. In Section 4.1.2, I am confused with the “human-judged ranking scores”, specifically how these scores are acquired. For example, Table 3 shows experiment results on Vicuna80 dataset. If I understood correctly, the human annotations are available in pairwise comparisons (as mentioned in Section 4.1.1). But it is unclear how these pairwise comparisons are converted into rankings of models?
    3. In the last line on page 10, it is mentioned that before discussions, GPT-4 is “not favoring GPT-3 much and is more fair” – does this contradict with the previously-mentioned self-enhancement bias? Or is it that GPT-4 only favors its own answers and not answers from previous versions such as GPT 3.

---

> ### Author Response · Authors · 2024-05-20
> **Rebuttal Content (1/2)**
>
> Thank you for your insightful comments and suggestions! We address your concerns below:
>
> ## Weakness 1: Certain technical descriptions (methods and results) lack support.
> We added the descriptions in the updated PDF.
>
> ## Weakness 2: clarity of writing
> >1. Authors mentioned “calculating metrics such as the win rate of each contestant” in Section 3.1, however, at this point, the audiences have no information of the answers and responses (until later in the experiment section) – what types (distributions) of questions were considered? Are they encouraged to be diverse? It might be good to provide some information here, so that the readers can connect the aforementioned “battle reviews” with the win rate matrices.
>
> We add a description of task questions-answers pairs, and battle reviews in Section 3. The added content is as follows:
>
> Specifically, the set of questions should cover various tasks, such as question answering (12.5%), email writing (12.5%), coding (6%), math solving (4%), etc. Answers should also vary in format, including concise answers, step-by-step reasonings, detailed explanations, code snippets, long-form answers, etc. Reviewers assess answer pairs and indicate preferences in the process (“battle review”). Then, both winrate and Elo metrics can be calculated.
>
> >2. In Section 4.1.2, I am confused with the “human-judged ranking scores”, specifically how these scores are acquired. For example, Table 3 shows experiment results on Vicuna80 dataset. If I understood correctly, the human annotations are available in pairwise comparisons (as mentioned in Section 4.1.1). But it is unclear how these pairwise comparisons are converted into rankings of models?
>
> Human-judged ranking scores are based on the win rate and Elo scores reported in Table 3, which are calculated based on human preferences by comparing each pair of model-generated answers. The dataset details include all the information. Moreover, Algorithms 1 and 2 describe the process of converting pairwise comparisons into rankings of models.
>
> >3. In the last line on page 10, it is mentioned that before discussions, GPT-4 is “not favoring GPT-3 much and is more fair” – does this contradict with the previously-mentioned self-enhancement bias? Or is it that GPT-4 only favors its own answers and not answers from previous versions such as GPT 3.
>
> In this claim, we didn’t mean GPT-4 has no self-enhancement bias. To avoid confusion, we removed “is more fair” from the paper. In section 4.3, we wanted to claim: ``Although GPT-4 still has the self-enhancement bias, it does not favor GPT-3’s answers.`` We fixed the writing in the PDF.
>
> ## Requested changes:
>
> >1. It seems that an intuitive way to mitigate this bias would be to put less weight for a model’s evaluation of its own answer. Could the author explain why this was not incorporated in the reviewer weight vector in PR? On a higher-level, I am wondering which procedure(s) in PR counter the self-enhancement bias.
>
> We add more details In section 3.1.1.
>
> Our Peer Rank method is a simple but effective way to automatically adjust weights for all models and mitigate the self-enhancement bias. During the automatic evaluation process, every reviewer model’s score is considered instead of only the model itself (Equation 2). While a reviewer may favor its own outputs, other reviewers’ scores provide a fair balance. Additionally, weaker models’ reviewing weights decrease automatically (to near zero) because of the normalization operation. The whole process mitigates self-enhancement bias. Empirical tests showed that fixing the self-weight at zero resulted in poorer performance.
>
> >2. I would suggest that author summarize the setting of Walsh (2014) in a more rigorous manner and explain why (or why not) the analysis in Walsh (2014) can extend to the setting studied in this paper
>
> We added the difference and our adapted proof in Appendix A.
>
> >3. How the proposed approach relates to prior works in more detail.
>
> We added more details In section 2, highlighted by blue text.
>
> Prior works on LLM-based discussion/debating (Liang et al. [1], Du et al. [2], Chan et al. [3]) focus more on improving **general task performance** (e.g., math reasoning and other NLP tasks). However, our work focuses on aligning with human preference. For example, on a question answering task (given questions as input and answers as output), prior works utilize LLM interactions to answer questions and improve models’ accuracy.  For our case, based on one response from humans and another from LLM, our approach utilizes LLM interactions to discuss which one is better and in order to **better align with human preferences**.
>
> Prior work on educational research mainly focuses on human-in-the-loop studies such as in the classroom (Cho & MacArthur, 2011; Nicol et al. (2014), Walsh, 2014). They conduct human-oriented data collection and experiments to verify the benefits of peer evaluations. While we focus on automatic evaluation.

---

> ### Author Response · Authors · 2024-05-20
> **Rebuttal Content (2/2)**
>
> >4. Should the variances across trials also be studied and reported?
>
> We added the variances in Table 7. ((±Numbers) represent standard deviations). The default decoding temperature of LLMs is 0.2. So, the variances are low.
>
> >5. Table 9 shows experiment results for PD, but the “contestant” model is GPT-3. I am curious to know why GPT-4 was not used as the answering model?
>
> We are conducting peer discussion (PD) in this section. PD does not involve pairwise battling among reviewers (as done in peer rank). Instead, only two evaluator LLMs hold discussions to achieve agreement and better evaluation results. We use the LFQA (long form question answering) dataset, it only contains GPT-3 answers and human answers.
>
> The PD discussions can be conducted upon answers from any models, such as GPT-4, GPT-3.5, Claude, PaLM-2, etc.
>
>
> >6. Statistical significance. In Table 7, I am confused about why there are two choices of p-values (0.05 and 0.005)
>
> They represent different degrees of significance, as is done in traditional NLP paper [4].
>
> >7. How many turns there are in the discussion. And whether the number of turns affect the accuracy of the evaluation outcome
>
> We added related details in section 4.1.3, highlighted by blue text.
>
> In our discussions, the maximum number of turns is 4. Models usually achieve mutual agreement within this limit. Experimenting with more turns yielded consistent results without altering the outcome.
>
> >8. “When we add to each turn’s prompt the role/identity information to remind the reviewer, the performance of GPT-4 leading discussions changes marginally, but Claude-leading PDA drops”. Without an explanation for this phenomenon, it is hard for readers to interpret a clear take-away message
>
> We updated the details In section 4.3.
>
> The last row (w/ role) in Table 5 on prompting’s effects was wrongly put in and they should correspond to the first row in Table 6. We fixed and denoted the change in blue.
>
> This original finding (claim) is invalid after the correction and we removed it from the PDF. The PDA scores are increased after adding the role information for both models in the prompt, which proves that the role information is helpful for LLMs in discussions.
>
> **References:**
> 1. Liang, Tian, et al. "Encouraging divergent thinking in large language models through multi-agent debate." arXiv preprint arXiv:2305.19118 (2023).
> 2. Du, Yilun, et al. "Improving factuality and reasoning in language models through multiagent debate." arXiv preprint arXiv:2305.14325 (2023).
> 3. Chan, Chi-Min et al. “ChatEval: Towards Better LLM-based Evaluators through Multi-Agent Debate.” ArXiv abs/2308.07201 (2023): n. Pag.
> 4. Yang, Bishan, Joint inference for fine-grained opinion extraction. ACL 2023.

---

> ### Comment · Reviewer_1dPG · 2024-06-05
> **Thank you for the response / some follow-up questions.**
>
> Thank you for your detailed response and for updating the paper accordingly. I have a couple of follow-up questions below.
>
> 1. In section 4.3, the texts for Table 8 state that “(1) when two reviewer LLMs are of similar capabilities (e.g., GPT-4 and Claude), there are likely relatively large improvements upon their initial reviews”, however such improvements only exist when GPT-4 is leading the discussion. For both columns “Random” and “R2-lead”, there appear to be no improvements upon GPT-4’s initial score 0.729. Does this mean that the slightly-stronger-reviewer needs to lead the discussion for the improvement to happen? I still have some doubts on this claim. Especially since it is hard to know in practice which model is the better one, so Random seems like the most reasonable option to compare to.
> 2. I appreciate the addition of variance of the results. How many trials were conducted to calculate the variance?
> 3. P-values. I am still confused why there are 2 different p-values. In other words, why are *different* degrees of significance needed?
> 4. It might be helpful to explicitly discuss the percentage of improvements and compare the improvements with the standard deviations in General Accuracy in Section 4.3.
> 5. In Section 3.1.1, the description of PR states that “Our win rate calculation assigns differing weight to the scores provided by different reviewers (A, B, C) based on the performance of the corresponding reviewers as a contestant (1, 2, 3).” – In the results, for example Table 3, I understand that All (Weighted) corresponds to this dynamic weighting. But then what exactly happens in All --- Is it simply an unweighted average of the scores given by the 5 reviewer models?
>
> Thanks.

---

> > ### Author Response · Authors · 2024-06-06
> > **Reply to follow-up questions**
> >
> > >1. In section 4.3, the texts for Table 8 state that “(1) when two reviewer LLMs are of similar capabilities (e.g., GPT-4 and Claude), there are likely relatively large improvements upon their initial reviews”, however such improvements only exist when GPT-4 is leading the discussion. For both columns “Random” and “R2-lead”, there appear to be no improvements upon GPT-4’s initial score 0.729. Does this mean that the slightly-stronger-reviewer needs to lead the discussion for the improvement to happen? I still have some doubts on this claim. Especially since it is hard to know in practice which model is the better one, so Random seems like the most reasonable option to compare to
> >
> > No. In Table 8, weaker models can also lead the discussion for improvements.
> > In row 2 and 3, the weaker model GPT-35 finally reaches the highest score.
> >
> > In practice, we can learn the power of models by our PR method and then use them to perform the PD task. Also, as you mentioned, we can apply the random strategy to improve the models’ performance.
> >
> > >2. I appreciate the addition of variance of the results. How many trials were conducted to calculate the variance?
> >
> > We run 5 trials to calculate the variance.
> >
> > >3. P-values. I am still confused why there are 2 different p-values. In other words, why are different degrees of significance needed?
> >
> > We unify them as p-value<0.05 in the PDF.
> >
> > >4. It might be helpful to explicitly discuss the percentage of improvements and compare the improvements with the standard deviations in General Accuracy in Section 4.3.
> >
> > We update the contents in Section 4.3 and add the percentage of improvements in paragraphs.
> >
> > >5. In Section 3.1.1, the description of PR states that “Our win rate calculation assigns differing weight to the scores provided by different reviewers (A, B, C) based on the performance of the corresponding reviewers as a contestant (1, 2, 3).” – In the results, for example Table 3, I understand that All (Weighted) corresponds to this dynamic weighting. But then what exactly happens in All --- Is it simply an unweighted average of the scores given by the 5 reviewer models?
> >
> > Yes

---

> > > ### Comment · Reviewer_1dPG · 2024-06-07
> > >
> > > Thank you for your response.

---

### Decision · Action_Editor_t7JX · 2024-06-28

**Recommendation:** Accept as is

**Comment:**

After three thorough reviews, discussions with authors, and appropriate revisions, we conclude that the manuscript supports all claims with suitable evidence. The topic is undoubtedly of interest to the community.

**Audience:**

Yes

**Claims And Evidence:**

Yes